# Embedding principle of homogeneous neural network for classification problem

**Jiahan Zhang**[1]**, Yaoyu Zhang**[1,2*]**, Tao Luo**[1,2,3†]

[1]School of Mathematical Sciences, Shanghai Jiao Tong University,
[2]Institute of Natural Sciences, MOE-LSC, Shanghai Jiao Tong University, Shanghai, 200240, China
[3]CMA-Shanghai, Shanghai Jiao Tong University, Shanghai, 200240, China

## Abstract

In this paper, we study the Karush-Kuhn-Tucker (KKT) points of the associated maximum-margin problem in homogeneous neural networks, including fully-connected and convolutional neural networks. In particular, We investigates the relationship between such KKT points across networks of different widths generated. We introduce and formalize the **KKT point embedding principle**, establishing that KKT points of a homogeneous network's max-margin problem $(P_\Phi)$ can be embedded into the KKT points of a larger network's problem $(P_{\tilde{\Phi}})$ via specific linear isometric transformations. We rigorously prove this principle holds for neuron splitting in fully-connected networks and channel splitting in convolutional neural networks. Furthermore, we connect this static embedding to the dynamics of gradient flow training with smooth losses. We demonstrate that trajectories initiated from appropriately mapped points remain mapped throughout training and that the resulting $\omega$-limit sets of directions are correspondingly mapped, thereby preserving the alignment with KKT directions dynamically when directional convergence occurs. We conduct several experiments to justify that trajectories are preserved. Our findings offer insights into the effects of network width, parameter redundancy, and the structural connections between solutions found via optimization in homogeneous networks of varying sizes.

## 1 Introduction

The optimization of neural networks remains a central challenge, with significant research dedicated to understanding the training dynamics and the properties of converged solutions. For homogeneous networks, such as those using ReLU-like activations without biases, a particularly fruitful line of inquiry connects optimization algorithms like gradient descent to the concept of margin maximization in classification tasks [Lyu and Li, 2019, Ji and Telgarsky, 2020]. In these settings, the Karush-Kuhn-Tucker (KKT) conditions associated with a minimum-norm maximum-margin problem provide a theoretical characterization of optimal parameter configurations [Gunasekar et al., 2018, Nacson et al., 2019b].

While much work focuses on the implicit bias and convergence properties within a single network architecture, a fundamental question arises regarding the relationship between solutions found in networks of different sizes. Specifically, how do the optimal solutions (characterized by KKT points) of a smaller homogeneous network relate to those of a larger network, particularly one derived by increasing width through operations like neuron splitting? Understanding this relationship is crucial

---

*Corresponding author: zhyy.sjtu@sjtu.edu.cn
†Corresponding author: luotao41@sjtu.edu.cn

39th Conference on Neural Information Processing Systems (NeurIPS 2025).

for insights into the effects of overparameterization, the structure of the solution space, and the mechanisms underlying implicit regularization.

To address this question, this paper introduces the **KKT Point Embedding Principle**. The core idea is that under specific, structure-preserving transformations corresponding to neuron splitting, the KKT points of the max-margin problem associated with a smaller homogeneous network ($\Phi$) can be precisely mapped, via a linear isometry $T$, to KKT points of the analogous problem for a larger network ($\tilde{\Phi}$). This principle provides a concrete link between the optimization landscapes and solution sets of related networks.

Specifically, our contributions include:

1. We formalize the KKT Point Embedding Principle (Theorem 4.2) and establishing conditions under which a linear transformation preserves KKT points between the max-margin problems $P_\Phi$ and $P_{\tilde{\Phi}}$(Theorem 4.5).

2. We provide explicit constructions and rigorous proofs demonstrating that this principle holds for neuron splitting in full-connected networks (Theorem 4.8) and channel splitting in convolutional neural networks(Theorem 4.11), showing these transformations satisfy the required conditions and are isometric.

3. We connect our static KKT embedding to the dynamics of training by proving a strong trajectory preservation principle. Specifically, we show that the entire gradient flow path of the wider network is a linear mapping of its narrower counterpart ($\boldsymbol{\eta}(t) = T\boldsymbol{\theta}(t)$) (Theorem 5.2). This dynamic correspondence directly implies that the asymptotic behavior is also preserved, ensuring the set of directional limit points is mapped accordingly ($T(L(\boldsymbol{\theta}(0))) = L(\boldsymbol{\eta}(0))$) and preserving the convergence towards KKT-aligned solutions (Theorem 5.4, Corollary 5.5).

The remainder of this paper is organized as follows. Section 2 discusses related work on implicit bias, network embeddings, and KKT conditions. Section 3 introduces necessary notations and definitions. Section 4 develops the static KKT Point Embedding Principle and applies it to neuron splitting and channl. Section 5 connects the static principle to gradient flow dynamics. Finally, Section 6 concludes the paper. An overview of our theoretical contributions and their relationships is provided in Figure 1.

## 2    Related work

Our work builds upon and contributes to several lines of research concerning neural network optimization, implicit bias, network structure, and optimality conditions.

**Optimization dynamics and implicit bias**    A significant body of research investigates the implicit bias of gradient-based optimization in neural networks. Seminal work by Soudry et al. [2018] showed that for linear logistic regression on separable data, gradient descent (GD) converges in direction to the $L^2$-max-margin solution. This finding was extended to various settings, including stochastic GD [Nacson et al., 2019c], and generalized to deep linear networks [Ji and Telgarsky, 2018]. For non-linear homogeneous neural networks (e.g., ReLU networks without biases), studies confirm that GD/GF with exponential-tailed losses also typically steer parameters towards margin maximization [Lyu and Li, 2019, Ji and Telgarsky, 2020, Nacson et al., 2019a, Wei et al., 2019]. The resulting solutions are frequently characterized as KKT points of an associated max-margin problem [Gunasekar et al., 2018, Nacson et al., 2019c].

More recently, this line of inquiry has been extended to analyze the margin in non-homogeneous models [Kunin et al., 2023, Cai et al., 2023] and to connect KKT conditions with the phenomenon of benign overfitting [Frei et al., 2023]. These analyses, however, primarily focus on the dynamics within a single, fixed-width network, with less exploration of how such KKT-characterized solutions relate across networks of varying widths, such as those generated by neuron splitting.

**Network embedding principles**    The idea that smaller network functionalities can be embedded within larger ones has been explored, notably by Zhang et al. [2021] who proposed an "Embedding Principle." This concept is related to foundational ideas on hierarchical structures and singularities

Figure 1: **Overview of Theoretical Contributions.** This figure illustrates the logical flow of our paper's theoretical framework. The analysis begins with the static principles (top section), which are then shown to apply to specific network architectures. These static results form the basis for the dynamic analysis (bottom row), which shows the preservation of training trajectories and their limit sets.

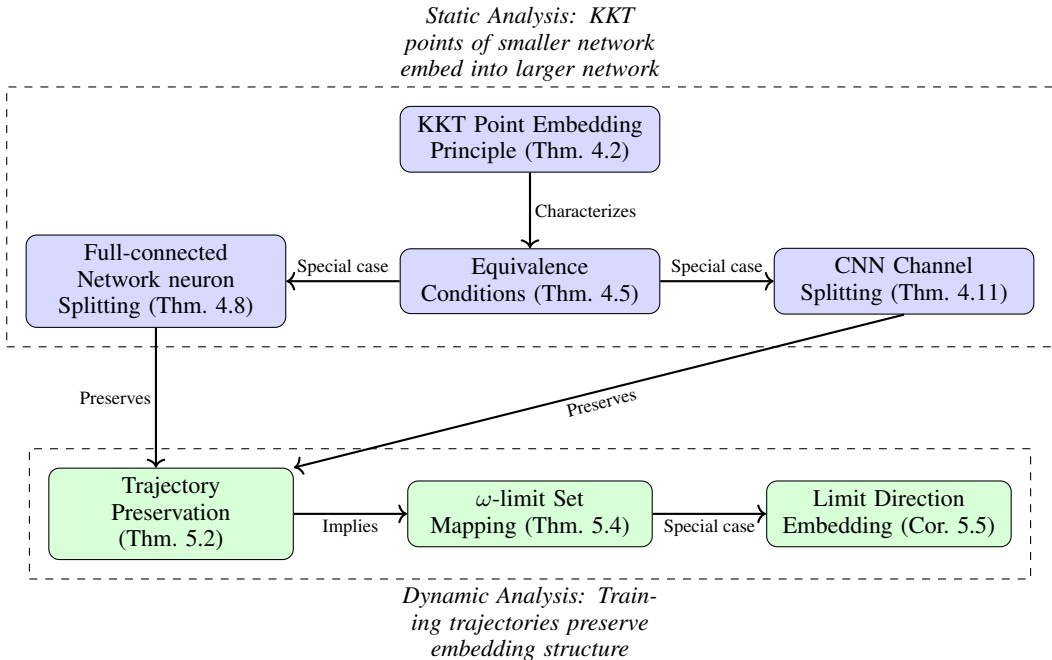

in the parameter space [Fukumizu and Amari, 2000], as well as more recent work exploring symmetries, such as permutation invariance, that connect different global minima [Simsek et al., 2021]. While general embedding principles focus on preserving the loss landscape, their specific application to the mapping of KKT points within max-margin settings, or to the detailed embedding of entire optimization trajectories, especially where parameter norms diverge, remains a less developed area warranting further investigation.

**KKT conditions in optimization and machine learning** The KKT conditions [Kuhn and Tucker, 1951] are fundamental for constrained optimization, extending to non-smooth problems via tools like the Clarke subdifferential [Clarke, 1975] (Definition 3.3), and famously characterizing SVM solutions [Cortes and Vapnik, 1995]. In deep learning, KKT conditions are crucial for theoretically characterizing network solutions. As highlighted, they are pivotal in understanding the implicit bias of GD towards max-margin solutions in homogeneous networks [Gunasekar et al., 2018, Nacson et al., 2019b]. These studies analyze KKT points of a specific minimum-norm, maximum-margin problem (like $P_\Phi$ in Definition 3.4) for a given network. While this establishes the nature of solutions within a fixed architecture, the structural relationship and potential embedding of these KKT points when transitioning between networks of different widths remains a complementary and important area of study.

## 3  Preliminaries

**Basic notations** For $n \in \mathbb{N}$, let $[n] := \{1, \ldots, n\}$. The Euclidean ($L^2$) norm of a vector $\boldsymbol{v}$ is denoted by $\|\boldsymbol{v}\|_2$. For a function $f : \mathbb{R}^d \to \mathbb{R}$, $\nabla f(\boldsymbol{x})$ is its gradient if $f$ is differentiable, and $\partial^\circ f(\boldsymbol{x})$ denotes its Clarke subdifferential [Clarke, 1975] if $f$ is locally Lipschitz (Definition A.1). A function $f : X \to \mathbb{R}^d$ is $\mathcal{C}^k$-smooth if it is $k$-times continuously differentiable, and $f : X \to \mathbb{R}$ is locally Lipschitz if it is Lipschitz continuous on a neighborhood of every point in $X$. Vectors are written in bold (e.g., $\boldsymbol{x}, \boldsymbol{\theta}$). Furthermore, for a linear map $T : \mathbb{R}^m \to \mathbb{R}^{\tilde{m}}$ and a set $S \subseteq \mathbb{R}^m$, we

denote the image of $S$ under $T$ as $TS := \{T\boldsymbol{x} \mid \boldsymbol{x} \in S\}$. Consequently, if $S = \partial^\circ f(\boldsymbol{x})$, its image is $T\partial^\circ f(\boldsymbol{x}) := \{T\boldsymbol{g} \mid \boldsymbol{g} \in \partial^\circ f(\boldsymbol{x})\}$.

**Definition 3.1** (Homogeneous neural network). *A neural network $\Phi(\boldsymbol{\theta}; \boldsymbol{x})$ with parameters $\boldsymbol{\theta}$ is (positively) homogeneous of order $L > 0$ if: for all $c > 0$: $\Phi(c\boldsymbol{\theta}; \boldsymbol{x}) = c^L \Phi(\boldsymbol{\theta}; \boldsymbol{x})$. Common examples include networks composed of linear layers and positive 1-homogeneous activations like ReLU ($\sigma(z) = \max(0, z)$), potentially excluding bias terms.*

**Binary classification setup**  We consider a dataset $\mathcal{D} = \{(\boldsymbol{x}_k, y_k)\}_{k=1}^n$, where $\boldsymbol{x}_k \in \mathbb{R}^{d_x}$ are input features and $y_k \in \{\pm 1\}$ are binary labels. The network $\Phi(\boldsymbol{\theta}; \boldsymbol{x})$ maps inputs to a scalar output, with parameters $\boldsymbol{\theta} \in \mathbb{R}^m$.

**Definition 3.2** (Margin). *The margin of the network $\Phi$ with parameters $\boldsymbol{\theta}$ on data point $k$ is $q_k(\boldsymbol{\theta}) := y_k \Phi(\boldsymbol{\theta}; \boldsymbol{x}_k)$. The margin on the dataset is $q_{\min}(\boldsymbol{\theta}) := \min_{k \in [n]} q_k(\boldsymbol{\theta})$. For an $L$-homogeneous network $\Phi$, the normalized margin is defined as $\bar{\gamma}(\boldsymbol{\theta}) := q_{\min}(\boldsymbol{\theta}) / \|\boldsymbol{\theta}\|_2^L$.*

**Definition 3.3** (KKT conditions for non-smooth problems). *Consider the optimization problem $\min_{\boldsymbol{x} \in \mathbb{R}^{d_x}} f(\boldsymbol{x})$ subject to $g_k(\boldsymbol{x}) \le 0$ for all $k \in [n]$, where $f$ and $g_k$ are locally Lipschitz functions. A feasible point $\boldsymbol{x}^*$ is a Karush-Kuhn-Tucker (KKT) point if there exist multipliers $\lambda_k \ge 0$, $k \in [n]$, such that:*

1. *Stationarity: $\mathbf{0} \in \partial^\circ f(\boldsymbol{x}^*) + \sum_{k=1}^n \lambda_k \partial^\circ g_k(\boldsymbol{x}^*)$.*

2. *Primal Feasibility: $g_k(\boldsymbol{x}^*) \le 0$, for all $k \in [n]$.*

3. *Dual Feasibility: $\lambda_k \ge 0$, for all $k \in [n]$.*

4. *Complementary Slackness: $\lambda_k g_k(\boldsymbol{x}^*) = 0$, for all $k \in [n]$.*

**Definition 3.4** (Minimum-norm max-margin problem $P_\Phi$). *Inspired by the implicit bias literature, we consider the problem of finding minimum-norm parameters that achieve a margin of at least 1:*

$$P_\Phi: \quad f(\boldsymbol{\theta}) = \min_{\boldsymbol{\theta} \in \mathbb{R}^m} \frac{1}{2} \|\boldsymbol{\theta}\|_2^2 \quad \text{subject to} \quad g_k(\boldsymbol{\theta}) := 1 - y_k \Phi(\boldsymbol{\theta}; \boldsymbol{x}_k) \le 0, \quad \text{for all } k \in [n].$$

*Here, the objective is $f(\boldsymbol{\theta}) = \frac{1}{2} \|\boldsymbol{\theta}\|_2^2$ (so $\partial^\circ f(\boldsymbol{\theta}) = \{\boldsymbol{\theta}\}$) and the constraints involve $g_k(\boldsymbol{\theta})$. Under Assumption A.3, $\partial^\circ g_k(\boldsymbol{\theta}) = -y_k \partial_{\boldsymbol{\theta}}^\circ \Phi(\boldsymbol{\theta}; \boldsymbol{x}_k)$. We denote the analogous problem for a different network $\tilde{\Phi}$ as $P_{\tilde{\Phi}}$. KKT points of this problem characterize directions associated with margin maximization.*

**Gradient flow**  We model the training dynamics using gradient flow (GF), which can be seen as gradient descent with infinitesimal step size. The parameter trajectory $\boldsymbol{\theta}(t)$ is an arc satisfying the differential inclusion $\frac{d\boldsymbol{\theta}(t)}{dt} \in -\partial^\circ \mathcal{L}(\boldsymbol{\theta}(t))$ for almost every $t \ge 0$. Here $\mathcal{L}(\boldsymbol{\theta}) := \sum_{k=1}^n \ell(y_k \Phi(\boldsymbol{\theta}; \boldsymbol{x}_k))$ is the training loss, where $\ell$ is a suitable loss function (e.g., exponential loss). We denote the analogous loss for network $\tilde{\Phi}$ as $\tilde{\mathcal{L}}$.

# 4 The KKT point embedding principle: Static setting

We first establish a general principle for mapping KKT points between constrained optimization problems linked by a linear transformation. We then specialize this principle to the context of homogeneous neural network classification via neuron splitting. Throughout this section, we operate under the following assumptions regarding the networks involved.

**Assumption 4.1** (Static setting). *Let $\Phi(\boldsymbol{\theta}; \boldsymbol{x})$ (parameters $\boldsymbol{\theta} \in \mathbb{R}^m$) and $\tilde{\Phi}(\boldsymbol{\eta}; \boldsymbol{x})$ (parameters $\boldsymbol{\eta} \in \mathbb{R}^{\tilde{m}}$) be two neural networks. We assume:*
*(A1) (Regularity) For any fixed input $\boldsymbol{x}$, the functions $\Phi(\cdot; \boldsymbol{x})$ and $\tilde{\Phi}(\cdot; \boldsymbol{x})$ mapping parameters to output are locally Lipschitz. Furthermore, they admit the application of subdifferential chain rules as needed (Assumption A.3).*
*(A2) (Homogeneity) Both $\Phi$ and $\tilde{\Phi}$ are positively homogeneous of the same order $L > 0$ with respect to their parameters (Definition 3.1).*

## 4.1   The KKT point embedding principle: a general framework

**Theorem 4.2** (General KKT mapping via linear transformation). *Let $f, g_k : \mathbb{R}^m \to \mathbb{R}$ and $\tilde{f}, \tilde{g}_k : \mathbb{R}^{\tilde{m}} \to \mathbb{R}$ be locally Lipschitz. Consider problems:*

$$(P) \quad \min f(\boldsymbol{\theta}) \quad s.t. \quad g_k(\boldsymbol{\theta}) \leq 0, \text{ for all } k \in [n].$$

$$(\tilde{P}) \quad \min \tilde{f}(\boldsymbol{\eta}) \quad s.t. \quad \tilde{g}_k(\boldsymbol{\eta}) \leq 0, \text{ for all } k \in [n].$$

*Let $T : \mathbb{R}^m \to \mathbb{R}^{\tilde{m}}$ be linear. Suppose:*

1. *Constraint Preserving: $\tilde{g}_k(T\boldsymbol{\theta}) = g_k(\boldsymbol{\theta})$, for all $\boldsymbol{\theta} \in \mathbb{R}^m$ and $k \in [n]$.*

2. *Objective Subgradient Preserving: $\partial^\circ \tilde{f}(T\boldsymbol{\theta}) = T\partial^\circ f(\boldsymbol{\theta})$, for all $\boldsymbol{\theta} \in \mathbb{R}^m$.*

3. *Constraint Subgradient Preserving: $\exists t_k > 0$ s.t. $\partial^\circ \tilde{g}_k(T\boldsymbol{\theta}) = t_k T\partial^\circ g_k(\boldsymbol{\theta})$, for all $\boldsymbol{\theta} \in \mathbb{R}^m$ and $k \in [n]$.*

*If $\boldsymbol{\theta}^*$ is a KKT point of $(P)$, then $\boldsymbol{\eta}^* = T\boldsymbol{\theta}^*$ is a KKT point of $(\tilde{P})$.*

*Proof.* Proof deferred to Appendix B.1.                                                     □

We now specialize Theorem 4.2 to the minimum-norm max-margin problem $P_\Phi$ (Definition 3.4).

**Definition 4.3** (KKT point preserving transformation). *Let $\Phi(\boldsymbol{\theta}; \boldsymbol{x})$ (with parameters $\boldsymbol{\theta} \in \mathbb{R}^m$) and $\tilde{\Phi}(\boldsymbol{\eta}; \boldsymbol{x})$ (with parameters $\boldsymbol{\eta} \in \mathbb{R}^l$) be two neural networks, and let $P_\Phi$ and $P_{\tilde{\Phi}}$ be their associated minimum-norm max-margin problems (as defined in Definition 3.4). A linear transformation $T : \mathbb{R}^m \to \mathbb{R}^l$ is called **KKT point preserving** from $P_\Phi$ to $P_{\tilde{\Phi}}$ if:*
*For any dataset $\mathcal{D} = \{(\boldsymbol{x}_k, y_k)\}_{k=1}^N$, if $\boldsymbol{\theta}^*$ is a KKT point of $P_\Phi$, then $\boldsymbol{\eta}^* = T(\boldsymbol{\theta}^*)$ is a KKT point of $P_{\tilde{\Phi}}$.*

**Proposition 4.4** (Composition of KKT point preserving transformations). *Let $\Phi(\boldsymbol{\theta}; \boldsymbol{x})$ (parameters $\boldsymbol{\theta} \in \mathbb{R}^m$), $\Phi_1(\boldsymbol{\eta}_1; \boldsymbol{x})$ (parameters $\boldsymbol{\eta}_1 \in \mathbb{R}^{m_1}$), and $\Phi_2(\boldsymbol{\eta}_2; \boldsymbol{x})$ (parameters $\boldsymbol{\eta}_2 \in \mathbb{R}^{m_2}$) be three neural networks, with inputs $\boldsymbol{x} \in \mathbb{R}^{d_x}$. Let $P_\Phi$, $P_{\Phi_1}$, and $P_{\Phi_2}$ be their respective associated minimum-norm max-margin problems (Definition 3.4).*

*Suppose $T_1 : \mathbb{R}^m \to \mathbb{R}^{m_1}$ is a linear transformation that is KKT Point Preserving from $P_\Phi$ to $P_{\Phi_1}$ (as per Definition 4.3). Suppose $T_2 : \mathbb{R}^{m_1} \to \mathbb{R}^{m_2}$ is a linear transformation that is KKT Point Preserving from $P_{\Phi_1}$ to $P_{\Phi_2}$ (as per Definition 4.3).*

*Then, the composite linear transformation $T = T_2 \circ T_1 : \mathbb{R}^m \to \mathbb{R}^{m_2}$ is KKT Point Preserving from $P_\Phi$ to $P_{\Phi_2}$. Furthermore, if $T_1$ and $T_2$ are isometries, then $T = T_2 \circ T_1$ is also an isometry.*

*Proof.* Proof deferred to Appendix B.2.                                                     □

**Theorem 4.5** (Equivalence conditions for KKT embedding). *Let $\Phi(\boldsymbol{\theta}; \boldsymbol{x})$ (parameters $\boldsymbol{\theta} \in \mathbb{R}^m$) and $\tilde{\Phi}(\boldsymbol{\eta}; \boldsymbol{x})$ (parameters $\boldsymbol{\eta} \in \mathbb{R}^l$) satisfy Assumptions 4.1 (A1, A2). Let $P_\Phi$ and $P_{\tilde{\Phi}}$ be their associated min-norm max-margin problems (Definition 3.4). Let $T : \mathbb{R}^m \to \mathbb{R}^l$ be a linear transformation. The following conditions are equivalent:*

1. *Output and subgradient preserving: for all $\boldsymbol{\theta} \in \mathbb{R}^m$ and $\boldsymbol{x} \in \mathbb{R}^{d_x}$,*

$$\tilde{\Phi}(T(\boldsymbol{\theta}); \boldsymbol{x}) = \Phi(\boldsymbol{\theta}; \boldsymbol{x}),$$
$$\exists \tau(\boldsymbol{\theta}, \boldsymbol{x}) > 0 \text{ s.t. } \partial_{\boldsymbol{\eta}}^\circ \tilde{\Phi}(T(\boldsymbol{\theta}); \boldsymbol{x}) = \tau(\boldsymbol{\theta}, \boldsymbol{x}) T(\partial_{\boldsymbol{\theta}}^\circ \Phi(\boldsymbol{\theta}; \boldsymbol{x})).$$

2. *The transformation $T$ is KKT Point Preserving from $P_\Phi$ to $P_{\tilde{\Phi}}$ (as per Definition 4.3).*

*Proof.* Proof deferred to Appendix B.3.                                                     □

## 4.2 Applications: KKT embedding via neuron and channel splitting

### 4.2.1 Neuron splitting in fully-connected networks

The principle extends naturally to splitting neurons in hidden layers of fully-connected homogeneous networks.

**Definition 4.6** (Neuron splitting transformation in fully-connected networks). *Let $\Phi$ be an $\alpha$-layer homogeneous network satisfying Assumptions 4.1(A1, A2), defined by weight matrices $W^{(l)}$ for $l \in [\alpha + 1]$ and positive 1-homogeneous activations $\sigma_l$ for hidden layers $l \in [\alpha]$. Let $\boldsymbol{\theta} = (\text{vec}(W^{(1)}), \ldots, \text{vec}(W^{(\alpha+1)})) \in \mathbb{R}^m$ be the parameter vector.*

*A neuron splitting transformation $T : \boldsymbol{\theta} \mapsto \boldsymbol{\eta}$ constructs a new network $\tilde{\Phi}$ (with parameters $\boldsymbol{\eta} \in \mathbb{R}^{\tilde{m}}$) from $\Phi$ by splitting a single neuron $j$ in a hidden layer $k$ ($1 \leq k \leq \alpha$) into $m_{split}$ new neurons (using $m_{split}$ to avoid clash with parameter dimension $m$). This splitting is defined by coefficients $c_i \geq 0$ for $i \in [m_{split}]$ such that $\sum_{i=1}^{m_{split}} c_i^2 = 1$. The transformation $T$ modifies the weights associated with the split neuron as follows:*

1. *The $j$-th row of $W^{(k)}$ (weights into neuron $j$, denoted $W_{j,:}^{(k)}$) is replaced by $m_{split}$ rows in the corresponding weight matrix $W'^{(k)}$ of $\tilde{\Phi}$. For each $i \in [m_{split}]$, the $i$-th of these new rows is $c_i W_{j,:}^{(k)}$.*

2. *The $j$-th column of $W^{(k+1)}$ (weights out of neuron $j$, denoted $W_{:,j}^{(k+1)}$) is replaced by $m_{split}$ columns in the corresponding weight matrix $W'^{(k+1)}$ of $\tilde{\Phi}$. For each $i \in [m_{split}]$, the $i$-th of these new columns is $c_i W_{:,j}^{(k+1)}$.*

3. *All other weights, rows, and columns in all weight matrices remain unchanged when mapped by $T$.*

*The resulting parameter vector for $\tilde{\Phi}$ is $\boldsymbol{\eta} = T\boldsymbol{\theta}$.*

**Remark 4.7** (Specialization to a two-layer network). *As a concrete example, the neuron splitting transformation in Definition 4.6, when applied to an $\alpha = 2$ layer network (i.e., a single hidden layer, $k = 1$), specializes to the two-layer neuron splitting described in Theorem A.4. Specifically, splitting hidden neuron $j$ involves transforming its input weights $W_{j,:}^{(1)}$ (analogous to $\boldsymbol{b}^\top$) to $m_{split}$ sets $c_i W_{j,:}^{(1)}$, and its output weights $W_{:,j}^{(2)}$ (analogous to $a$) to $m_{split}$ sets $c_i W_{:,j}^{(2)}$. This results in effective parameters $(c_i a, c_i \boldsymbol{b})$ for each $i$-th split part of the neuron, consistent with Theorem A.4.*

**Theorem 4.8** (Properties of deep neuron splitting transformation and KKT embedding). *Let $T$ be the deep neuron splitting transformation defined in Definition 4.6. Let $P_\Phi$ and $P_{\tilde{\Phi}}$ be the min-norm max-margin problems (Definition 3.4) associated with the original network $\Phi$ (with parameters $\boldsymbol{\theta} \in \mathbb{R}^m$) and the split network $\tilde{\Phi}$ (with parameters $\boldsymbol{\eta} \in \mathbb{R}^{\tilde{m}}$), respectively. Input data $\boldsymbol{x} \in \mathbb{R}^{d_x}$. The transformation $T$ satisfies the following properties:*

1. *$T$ is a linear isometry: $\|T\boldsymbol{\theta}\|_2 = \|\boldsymbol{\theta}\|_2$, for all $\boldsymbol{\theta} \in \mathbb{R}^m$.*

2. *Output preserving: $\tilde{\Phi}(T\boldsymbol{\theta}; \boldsymbol{x}) = \Phi(\boldsymbol{\theta}; \boldsymbol{x})$ for all $\boldsymbol{\theta} \in \mathbb{R}^m$ and $\boldsymbol{x} \in \mathbb{R}^{d_x}$.*

3. *Subgradient preserving: $\partial_{\boldsymbol{\eta}}^\circ \tilde{\Phi}(T\boldsymbol{\theta}; \boldsymbol{x}) = T(\partial_{\boldsymbol{\theta}}^\circ \Phi(\boldsymbol{\theta}; \boldsymbol{x}))$ for all $\boldsymbol{\theta} \in \mathbb{R}^m$ and $\boldsymbol{x} \in \mathbb{R}^{d_x}$. (This implies $\tau(\boldsymbol{\theta}, \boldsymbol{x}) = 1$ in the context of Theorem 4.5).*

*Consequently, since $T$ satisfies the conditions (specifically, condition 1 with $\tau = 1$) of Theorem 4.5, this deep neuron splitting transformation $T$ is a KKT point preserving from $P_\Phi$ to $P_{\tilde{\Phi}}$.*

*Proof.* The proof proceeds by verifying the three listed properties of $T$. Isometry (1) follows from comparing the squared Euclidean norms of the parameter vectors. Output Preservation (2) is demonstrated by tracing the signal propagation through the forward pass of both networks. Subgradient Equality (3) is established via a careful analysis of the backward pass using chain rules for Clarke subdifferentials. The preservation of KKT points is then a direct consequence of Theorem 4.5, given that $T$ fulfills its required conditions. The complete proof of properties (1) - (3) is provided in Appendix B.4. $\qquad\square$

**Remark 4.9** (Iterative splitting and KKT preservation). *Theorem 4.8 establishes that a single deep neuron splitting operation (Definition 4.6) is KKT Point Preserving (Definition 4.3). Since the composition of KKT Point Preserving transformations also yields a KKT Point Preserving transformation (Proposition 4.4), it follows directly that any finite sequence of such deep neuron splitting operations results in a composite transformation that is KKT Point Preserving.*

### 4.2.2 Channel splitting in convolutional neural networks

To demonstrate the generality of our framework beyond fully-connected architectures, we now extend the KKT Point Embedding Principle to Convolutional Neural Networks (CNNs). Applying our principle to CNNs presents a unique challenge: the transformation must respect the architectural hallmarks of convolutions, namely weight sharing and locality. We address this by designing a novel *channel splitting* transformation that preserves the network function and subgradient structure, thereby satisfying the conditions of Theorem 4.5.

**Definition 4.10** (Channel splitting transformation in deep CNNs). *Let $\Phi$ be a deep homogeneous CNN satisfying Assumptions 4.1(A1, A2), defined by a sequence of convolutional filters (weights) $\{W^{(l)}\}$. A **channel splitting transformation** $T : \boldsymbol{\theta} \mapsto \boldsymbol{\eta}$ constructs a new network $\tilde{\Phi}$ from $\Phi$ by splitting a single output channel $j$ of a hidden convolutional layer $k$ into $m_{split}$ new channels.*

*This transformation is defined by a set of coefficients $c_i \geq 0$ for $i \in [m_{split}]$ that satisfy the isometric condition $\sum_{i=1}^{m_{split}} c_i^2 = 1$. The transformation modifies the filters associated with the split channel as follows:*

- ***At Layer k***: *The filter $W_{j,:}^{(k)}$ that produces output channel $j$ is replaced by $m_{split}$ new filters in the corresponding weight tensor $\tilde{W}^{(k)}$ of $\tilde{\Phi}$. For each $i \in [m_{split}]$, the $i$-th of these new filters is defined as $c_i W_{j,:}^{(k)}$.*

- ***At Layer k+1***: *In every filter in the subsequent layer, $W^{(k+1)}$, the input slice corresponding to the original channel $j$ is replaced by $m_{split}$ new input slices. For each $i \in [m_{split}]$, the $i$-th of these new input slices is scaled by the corresponding coefficient $c_i$.*

- ***Other Weights***: *All other filters and weights in the network remain unchanged when mapped by $T$.*

*The resulting parameter vector for $\tilde{\Phi}$ is $\boldsymbol{\eta} = T\boldsymbol{\theta}$.*

This meticulously constructed transformation preserves the functional output of the network while allowing for an isometric embedding of the parameter space. We now state the main result for this section, which shows that this transformation satisfies our core theory.

**Theorem 4.11** (Properties of CNN channel splitting and KKT embedding). *Let $T$ be the deep CNN channel splitting transformation defined in Definition 4.10. Let $P_\Phi$ and $P_{\tilde{\Phi}}$ be the min-norm max-margin problems associated with the original network $\Phi$ and the split network $\tilde{\Phi}$, respectively. The transformation $T$ satisfies the following properties:*

1. *$T$ is a linear isometry: $\|T\boldsymbol{\theta}\|_2 = \|\boldsymbol{\theta}\|_2$ for all $\boldsymbol{\theta}$.*

2. *Output preserving: $\tilde{\Phi}(T\boldsymbol{\theta}; x) = \Phi(\boldsymbol{\theta}; x)$ for all $\boldsymbol{\theta}$ and input $x$.*

3. *Subgradient preserving: $\partial_{\boldsymbol{\eta}}^\circ \tilde{\Phi}(T\boldsymbol{\theta}; x) = T(\partial_{\boldsymbol{\theta}}^\circ \Phi(\boldsymbol{\theta}; x))$ for all $\boldsymbol{\theta}$ and input $x$.*

*Consequently, since $T$ satisfies the conditions of Theorem 4.5, this channel splitting transformation is a KKT point preserving from $P_\Phi$ to $P_{\tilde{\Phi}}$.*

*Proof.* The proof is analogous to the one for fully-connected networks (Theorem 4.8) and proceeds by verifying the three listed properties of $T$. Isometry (1) is confirmed by comparing the squared Frobenius norms of the filter tensors. Output Preservation (2) is demonstrated by tracing the feature map computations through the forward pass, showing that the split signals perfectly recombine at layer $k + 1$. Subgradient Preservation (3) is established by applying the chain rule for Clarke subdifferentials to the backward pass. The complete mathematical details are provided in Appendix B.5. □

The successful extension of our principle to CNNs validates our framework as a flexible blueprint applicable to a broad class of homogeneous networks.

# 5 Connection to training dynamics via gradient flow

Having established the static KKT Point Embedding Principle, we now investigate its implications for the dynamics of training homogeneous networks using gradient flow. We focus on the scenario where gradient flow converges towards max-margin solutions, a phenomenon linked to specific loss functions. We connect the parameter trajectories and limit directions of the smaller network $\Phi$ and the larger network $\tilde{\Phi}$ related by the neuron splitting transformation $T$.

We analyze the asymptotic directional behavior using the concept of the $\omega$-limit set from dynamical systems theory. Recall that for a trajectory $\mathbf{z}(t)$ evolving in some space, its $\omega$-limit set, denoted $\omega(\mathbf{z}_0)$ (where $\mathbf{z}_0$ is the initial point), is the set of all points $\mathbf{y}$ such that $\mathbf{z}(t_k) \to \mathbf{y}$ for some sequence of times $t_k \to \infty$. Intuitively, it's the set of points the trajectory approaches infinitely often as $t \to \infty$.

For our dynamic setting, we make the following assumptions, building upon the static ones (A1, A2 from Assumption 4.1):

**Assumption 5.1** (Dynamic setting). *In addition to Assumptions 4.1(A1, A2), we assume:*
*(A3) (Loss Smoothness) The per-sample loss function $\ell : \mathbb{R} \to \mathbb{R}$ is $\mathcal{C}^1$-smooth and non-increasing.*

Building on this, we first demonstrate that the neuron splitting transformation $T$ preserves the gradient flow trajectory itself. This result relies only on the smoothness of the loss and the network properties.

**Theorem 5.2** (Trajectory preserving for neuron splitting). *Let $\Phi, \tilde{\Phi}$ be homogeneous networks related by a neuron splitting transformation $T$ (as defined in Theorem A.4 or 4.8), satisfying Assumptions 4.1(A1, A2) and 5.1(A3). Consider the gradient flow dynamics for the respective losses $\mathcal{L}(\boldsymbol{\theta})$ and $\tilde{\mathcal{L}}(\boldsymbol{\eta})$. If the initial conditions are related by $\boldsymbol{\eta}(0) = T\boldsymbol{\theta}(0)$, then the trajectories satisfy $\boldsymbol{\eta}(t) = T\boldsymbol{\theta}(t)$ for all $t \geq 0$ (assuming solutions exist and norms diverge for normalization where needed, e.g., as per an implicit Assumption (A4) mentioned in dependent definitions/theorems).*

*Proof.* The proof relies on showing that the subdifferential of the loss function transforms according to $T(\partial^{\circ}\mathcal{L}(\boldsymbol{\theta})) = \partial^{\circ}\tilde{\mathcal{L}}(T\boldsymbol{\theta})$, which follows from the output Preservation and subgradient equality properties of $T$ (verified in Theorems A.4, 4.8) and the chain rule applied with the $\mathcal{C}^1$-smooth loss $\ell$ (Assumption A3 from 5.1). Full details are in Appendix C.1. □

This trajectory mapping allows us to relate the asymptotic directional behavior. We first define the set of directional limit points.

**Definition 5.3** ($\omega$-limit set of the normalized trajectory). *For a given gradient flow trajectory $\boldsymbol{\theta}(t)$ starting from $\boldsymbol{\theta}(0)$ (with $\boldsymbol{\theta} \in \mathbb{R}^m$) under Assumptions 4.1(A1, A2) and 5.1(A3) (and implicitly assuming trajectory properties like norm divergence for normalization, often denoted as (A4) in related theorems), let $\bar{\boldsymbol{\theta}}(t) = \boldsymbol{\theta}(t)/\|\boldsymbol{\theta}(t)\|_2$ be the normalized trajectory. The $\omega$-limit set [see, e.g., Hirsch et al., 2013, for the general theory and properties] of this normalized trajectory $\bar{\boldsymbol{\theta}}(t)$ is defined as*

$$\omega(\bar{\boldsymbol{\theta}}) := \left\{ \boldsymbol{x} \in \mathbb{S}^{m-1} \mid \exists \{t_k\}_{k=1}^{\infty} \text{ s.t. } t_k \to \infty \text{ and } \bar{\boldsymbol{\theta}}(t_k) \to \boldsymbol{x} \text{ as } k \to \infty \right\},$$

*where $\mathbb{S}^{m-1}$ is the unit sphere in the parameter space $\mathbb{R}^m$ of $\boldsymbol{\theta}$. This set $\omega(\bar{\boldsymbol{\theta}})$ contains all directional accumulation points of the trajectory $\boldsymbol{\theta}(t)$. Since $\bar{\boldsymbol{\theta}}(t)$ lies in the compact set $\mathbb{S}^{m-1}$, $L(\boldsymbol{\theta}(0))$ is non-empty and compact. If the direction $\bar{\boldsymbol{\theta}}(t)$ converges to a unique limit $\bar{\boldsymbol{\theta}}^*$, then $\omega(\bar{\boldsymbol{\theta}}) = \{\bar{\boldsymbol{\theta}}^*\}$.*

The following theorem shows that the transformation $T$ provides an exact mapping between the $\omega$-limit sets of the normalized trajectories. This relies on the trajectory mapping (Theorem 5.2) and the properties of $T$.

**Theorem 5.4** (Mapping of $\omega$-limit sets of normalized trajectories). *Let $\Phi, \tilde{\Phi}$ be homogeneous networks related by a neuron splitting transformation $T$ (Theorem A.4 or 4.8), satisfying Assumptions 4.1(A1, A2), Assumption 5.1(A3), and further assuming trajectory properties (A4: unique*

*solution existence and norm divergence for $\boldsymbol{\theta}(t)$ when data is classifiable). Let $\boldsymbol{\theta}(t)$ (in $\mathbb{R}^m$) and $\boldsymbol{\eta}(t)$ (in $\mathbb{R}^{\tilde{m}}$) be gradient flow trajectories starting from $\boldsymbol{\theta}(0)$ and $\boldsymbol{\eta}(0) = T\boldsymbol{\theta}(0)$ respectively. Let $L(\boldsymbol{\theta}(0)) \subseteq \mathbb{S}^{m-1}$ and $L(\boldsymbol{\eta}(0)) \subseteq \mathbb{S}^{l-1}$ be the $\omega$-limit sets of the respective normalized trajectories (Definition 5.3). Then, these sets are related by $T$:*

$$T(L(\boldsymbol{\theta}(0))) = L(\boldsymbol{\eta}(0))$$

*where $T(L(\boldsymbol{\theta}(0))) = \{T\boldsymbol{x} \mid \boldsymbol{x} \in L(\boldsymbol{\theta}(0))\}$.*

*Proof.* The proof shows inclusions in both directions using the continuity of $T$, the trajectory mapping $\bar{\boldsymbol{\eta}}(t) = T(\bar{\boldsymbol{\theta}}(t))$ (derived from Theorem 5.2 and isometry of $T$), and compactness arguments. It relies on Assumption (A4) for the existence and divergence of trajectories needed for normalization. The proof does not require the uniqueness of limits. The full proof is in Appendix C.2. $\square$

**Corollary 5.5** (Embedding of unique limit directions). *Let $\Phi, \tilde{\Phi}$ be homogeneous networks related by a neuron splitting transformation $T$ (Theorem A.4 or 4.8), satisfying Assumptions 4.1(A1, A2), Assumption 5.1(A3). Let $\boldsymbol{\theta}(t)$ and $\boldsymbol{\eta}(t)$ be gradient flow trajectories starting from $\boldsymbol{\theta}(0)$ and $\boldsymbol{\eta}(0) = T\boldsymbol{\theta}(0)$ respectively. If the normalized trajectory $\bar{\boldsymbol{\theta}}(t) = \boldsymbol{\theta}(t)/\|\boldsymbol{\theta}(t)\|_2$ converges to a unique limit direction $\bar{\boldsymbol{\theta}}^*$ as $t \to \infty$, then the corresponding normalized trajectory $\bar{\boldsymbol{\eta}}(t) = \boldsymbol{\eta}(t)/\|\boldsymbol{\eta}(t)\|_2$ converges to the unique limit direction $T\bar{\boldsymbol{\theta}}^*$.*

*Proof.* Proof deferred to Appendix C.3. $\square$

Corollary 5.5, which shows the embedding of unique limit directions ($T\bar{\boldsymbol{\theta}}^*$), is particularly relevant given the optimization dynamics observed in homogeneous neural networks. As established in prior work (e.g., [Lyu and Li, 2019, Ji and Telgarsky, 2020]), when training such homogeneous models (including fully-connected and convolutional networks with ReLU-like activations) with common classification losses such as logistic or cross-entropy loss, methods like gradient descent or gradient flow often steer the parameter direction $\bar{\boldsymbol{\theta}}(t)$ to align with solutions that maximize the (normalized) classification margin. These margin-maximizing directions are typically characterized as KKT points of an associated constrained optimization problem, akin to $P_\Phi$ discussed earlier (Definition 3.4). Therefore, our corollary, by demonstrating that the transformation $T$ preserves unique limit directions, implies that this neuron splitting mechanism effectively embeds the structure of these significant, KKT-aligned, margin-maximizing directions from a smaller network into a larger one. This preservation of margin-related properties through embedding is also noteworthy due to the well-recognized connection between classification margin and model robustness.

**Empirical Validation and Implications** [3]    A key prediction of our theoretical results is the principle of trajectory preservation (Theorem 5.2). To justify this claim empirically, we conducted experiments verifying this principle using discrete-time gradient descent. We trained pairs of narrow and wide homogeneous MLPs on a 2D linearly separable dataset (Exp. 1), with full implementation details deferred to Appendix D.

Figure 2 presents the results. As shown in the left panel of Figure 2, the trajectory error, $\|\boldsymbol{\eta}(t) - T\boldsymbol{\theta}(t)\|_2$, remains at the level of machine precision ($\sim 10^{-13}$) throughout training. Concurrently, the right panel shows the training loss converging towards zero, indicating a successful learning process. These results provide strong empirical validation for our theoretical claim under standard GD. Appendix D summarizes further experiments demonstrating robustness to SGD (with identical batch sequences) and non-separable data.

A direct practical implication of trajectory preservation is that the function learned by the wider network, $\tilde{\Phi}$, is identical to that of its narrower counterpart, $\Phi$, at every training step ($\tilde{\Phi}(\boldsymbol{\eta}(t); \cdot) = \Phi(\boldsymbol{\theta}(t); \cdot)$). This suggests that widening a network via our proposed splitting transformation can create significant parameter redundancy without altering the optimization path in function space. This result empirically supports the idea that solutions from simpler models are structurally embedded within their overparameterized counterparts.

---

[3]Code available: `https://github.com/Silentmoonlight/kkt-embedding-principle`

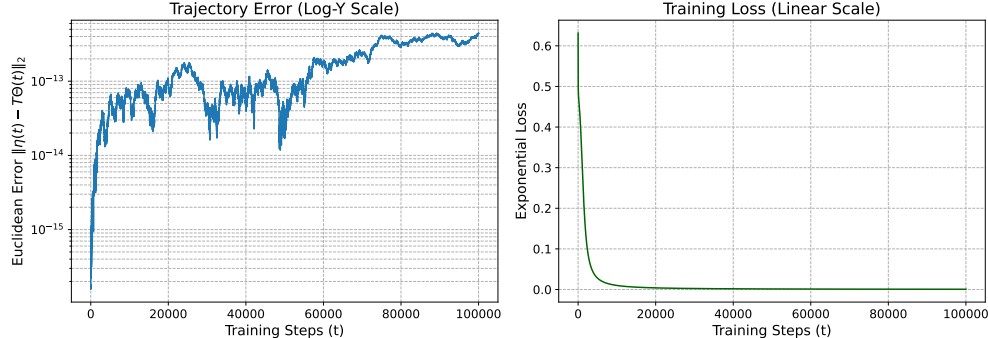

Figure 2: Empirical validation using MLP with GD on 2D toy data (Exp. 1). **(Left)** The trajectory error remains near machine precision throughout training. **(Right)** The training loss converges towards zero, indicating successful training.

## 6 Conclusion

This paper introduced the KKT Point Embedding Principle for homogeneous neural networks in max-margin classification. We established that specific neuron splitting transformations $T$, which are linear isometries, map KKT points of the associated min-norm problem $P_\Phi$ for a smaller network to those of an augmented network $P_{\tilde{\Phi}}$ (Theorems 4.5, 4.8, 4.11.) This static embedding was then connected to training dynamics: we proved that $T$ preserves gradient flow trajectories and maps the $\omega$-limit sets of parameter directions ($T(L(\boldsymbol{\theta}(0))) = L(\boldsymbol{\eta}(0))$) (Theorems 5.2, 5.4). Consequently, this framework allows for the dynamic preservation of KKT directional alignment through the embedding when such convergence occurs

Our work establishes a foundational KKT point embedding principle and its dynamic implications, leading to several natural further questions:

- What are the conditions for transformations $T$ to preserve KKT point structures in other homogeneous network architectures, such as Convolutional Neural Networks (CNNs)? This would likely involve designing transformations $T$ that respect architectural specificities (e.g., locality, weight sharing in CNNs) while satisfying the necessary isometric and output/subgradient mapping properties identified in our current work.

- Can the KKT Point Embedding Principle be generalized to non-homogeneous neural networks, for instance, those incorporating bias terms or employing activation functions that are not positively 1-homogeneous? This presents a significant challenge as homogeneity is central to the current analysis, and new theoretical approaches might be required to define and analyze analogous embedding phenomena.

## Acknowledgments and Disclosure of Funding

This work is sponsored by the National Key R&D Program of China (Grant No. 2022YFA1008200, T. L., Y. Z.), the Natural Science Foundation of China (No. 12571567, Y. Z.), and the Natural Science Foundation of Shanghai (No. 25ZR1402280, Y. Z.). We also thank Shanghai Institute for Mathematics and Interdisciplinary Sciences (SIMIS) for their financial support. This research was funded by SIMIS under grant number SIMIS-ID-2025-ST (T. L.). The authors are grateful for the resources and facilities provided by SIMIS, which were essential for the completion of this work.

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

# A    Mathematical preliminaries and illustrative example

**Definition A.1** (Clarke's subdifferential). *For a locally Lipschitz function $f : X \to \mathbb{R}$, the Clarke subdifferential[Clarke, 1975] at $\boldsymbol{x} \in X$ is*

$$\partial^\circ f(\boldsymbol{x}) \coloneqq \mathrm{conv} \left\{ \lim_{k \to \infty} \nabla f(\boldsymbol{x}_k) : \boldsymbol{x}_k \to \boldsymbol{x}, f \text{ is differentiable at } \boldsymbol{x}_k \right\},$$

*where* $\mathrm{conv}$ *denotes the convex hull. If $f$ is $\mathcal{C}^1$ near $\boldsymbol{x}$, $\partial^\circ f(\boldsymbol{x}) = \{\nabla f(\boldsymbol{x})\}$.*

**Definition A.2** (Arc and admissible chain rule). *We say that a function $z : I \to \mathbb{R}^d$ on the interval $I$ is an **arc** if $z$ is absolutely continuous for any compact sub-interval of $I$. For an arc $z$, $\dot{z}(t)$ (or $\frac{d}{dt}z(t)$) stands for the derivative at $t$ if it exists. Following the terminology in Davis et al. [2020], we say that a locally Lipschitz function $f : \mathbb{R}^d \to \mathbb{R}$ **admits a chain rule** if for any arc $z : [0, +\infty) \to \mathbb{R}^d$, for all $h \in \partial^\circ f$, the equality $(f \circ z)'(t) = \langle h, \dot{z}(t) \rangle$ holds for a.e. $t > 0$.*

**Assumption A.3** (Admissibility of subdifferential chain rule). *We assume throughout that the neural networks and loss functions involved are such that standard chain rules for Clarke subdifferentials apply as needed for backpropagation. Specifically, for compositions like $\mathcal{L}(\boldsymbol{\theta}) = \sum_{k=1}^{n} \ell(y_k \Phi(\boldsymbol{\theta}; \boldsymbol{x}_k))$, we assume the subdifferential $\partial^\circ \mathcal{L}(\boldsymbol{\theta})$ can be computed by propagating subgradients layer-wise. Conditions ensuring this are discussed in, e.g., Davis et al. [2020], Bolte and Pauwels [2021].*

**Theorem A.4** (Two-layer neuron splitting preserves KKT points). *Consider a single hidden neuron network $\Phi(\boldsymbol{\theta}; \boldsymbol{x}) = a\sigma(\boldsymbol{b}^\top \boldsymbol{x})$ with parameters $\boldsymbol{\theta} = (a, \boldsymbol{b}^\top) \in \mathbb{R}^{1+d_x}$ (where $a \in \mathbb{R}, \boldsymbol{b} \in \mathbb{R}^{d_x}$), and $\sigma$ is a positive 1-homogeneous activation function (e.g., ReLU) satisfying Assumption 4.1(A1). Consider a network $\tilde{\Phi}(\boldsymbol{\eta}; \boldsymbol{x}) = \sum_{i=1}^{k} a_i \sigma(\boldsymbol{b}_i^\top \boldsymbol{x})$ with parameters $\boldsymbol{\eta} = (a_1, \ldots, a_k, \boldsymbol{b}_1^\top, \ldots, \boldsymbol{b}_k^\top) \in \mathbb{R}^{k(1+d_x)}$. Define the linear transformation $T : \mathbb{R}^{1+d_x} \to \mathbb{R}^{k(1+d_x)}$ by $T(a, \boldsymbol{b}^\top) = (c_1 a, \ldots, c_k a, c_1 \boldsymbol{b}^\top, \ldots, c_k \boldsymbol{b}^\top)$, where $c_i \geq 0$ are splitting coefficients satisfying $\sum_{i=1}^{k} c_i^2 = 1$. Then $T$ satisfies:*

1. *Output preserving: $\tilde{\Phi}(T\boldsymbol{\theta}; \boldsymbol{x}) = \Phi(\boldsymbol{\theta}; \boldsymbol{x})$, for all $\boldsymbol{\theta} \in \mathbb{R}^{1+d_x}$ and $\boldsymbol{x} \in \mathbb{R}^{d_x}$.*

2. *Subgradient preserving: $\partial^\circ_{\boldsymbol{\eta}} \tilde{\Phi}(T\boldsymbol{\theta}; \boldsymbol{x}) = T(\partial^\circ_{\boldsymbol{\theta}} \Phi(\boldsymbol{\theta}; \boldsymbol{x}))$, for all $\boldsymbol{\theta} \in \mathbb{R}^{1+d_x}$ and $\boldsymbol{x} \in \mathbb{R}^{d_x}$.*

3. *Isometry: $T$ is a linear isometry ($\|T\boldsymbol{\theta}\|_2 = \|\boldsymbol{\theta}\|_2$ for all $\boldsymbol{\theta} \in \mathbb{R}^{1+d_x}$).*

*Consequently, by Theorem 4.5, this neuron splitting transformation $T$ is a KKT point preserving from $P_\Phi$ and $P_{\tilde{\Phi}}$.*

*Proof.* The proof involves directly verifying the three properties using the definitions of $\Phi$, $\tilde{\Phi}$, $T$, and subdifferential calculus. Let $\boldsymbol{\theta} = (a, \boldsymbol{b}^\top)^\top$.

**1. Output Preservation**: We compute $\tilde{\Phi}$ at $\boldsymbol{\eta} = T\boldsymbol{\theta}$:

$$\tilde{\Phi}(T\boldsymbol{\theta}; \boldsymbol{x}) = \sum_{i=1}^{k} a_i \sigma(\boldsymbol{b}_i^\top \boldsymbol{x})$$

$$= \sum_{i=1}^{k} (c_i a)\sigma((c_i \boldsymbol{b})^\top \boldsymbol{x}) \qquad \text{(Substituting transformed parameters)}$$

$$= \sum_{i=1}^{k} (c_i a)\sigma(c_i(\boldsymbol{b}^\top \boldsymbol{x}))$$

$$= \sum_{i=1}^{k} (c_i a)(c_i \sigma(\boldsymbol{b}^\top \boldsymbol{x})) \qquad \text{(Using 1-homogeneity of } \sigma)$$

$$= \left( \sum_{i=1}^{k} c_i^2 \right) a\sigma(\boldsymbol{b}^\top \boldsymbol{x})$$

$$= 1 \cdot a\sigma(\boldsymbol{b}^\top \boldsymbol{x}) \qquad \text{(Since } \sum c_i^2 = 1)$$

$$= \Phi(\boldsymbol{\theta}; \boldsymbol{x})$$

Thus, the network output is preserved.

**2. Subgradient Preservation**: Let $z = \boldsymbol{b}^\top \boldsymbol{x}$. The Clarke subdifferential of $\Phi$ w.r.t. $\boldsymbol{\theta}$ is $\partial_{\boldsymbol{\theta}}^\circ \Phi(\boldsymbol{\theta}; \boldsymbol{x}) = (\partial_a^\circ \Phi, \partial_{\boldsymbol{b}}^\circ \Phi)$, where $\partial_a^\circ \Phi = \{\sigma(z)\}$ and $\partial_{\boldsymbol{b}}^\circ \Phi = \{a \cdot g \cdot \boldsymbol{x} \mid g \in \partial^\circ \sigma(z)\}$. Applying $T$ to an element of $\partial_{\boldsymbol{\theta}}^\circ \Phi(\boldsymbol{\theta}; \boldsymbol{x})$ yields a vector with components corresponding to $(c_1 \sigma(z), \ldots, c_k \sigma(z))$ for the 'a' parts and $(c_1 a g \boldsymbol{x}, \ldots, c_k a g \boldsymbol{x})$ for the 'b' parts. Next, we compute the subdifferential of $\tilde{\Phi}$ at $\boldsymbol{\eta} = T\boldsymbol{\theta}$. For each component $j \in [k]$:

- $\partial_{a_j}^\circ \tilde{\Phi} = \{\sigma(\boldsymbol{b}_j^\top \boldsymbol{x})\} = \{\sigma(c_j z)\} = \{c_j \sigma(z)\}$

- $\partial_{\boldsymbol{b}_j}^\circ \tilde{\Phi} = \{a_j \cdot g_j \cdot \boldsymbol{x} \mid g_j \in \partial^\circ \sigma(\boldsymbol{b}_j^\top \boldsymbol{x})\} = \{c_j a \cdot g \cdot \boldsymbol{x} \mid g \in \partial^\circ \sigma(z)\}$

The equalities follow from the 1-homogeneity of $\sigma$ and the 0-homogeneity of its subdifferential $\partial^\circ \sigma$. Assembling these partials for a chosen $g \in \partial^\circ \sigma(z)$ shows that any element of $\partial_{\boldsymbol{\eta}}^\circ \tilde{\Phi}(T\boldsymbol{\theta}; \boldsymbol{x})$ matches the structure of an element in $T(\partial_{\boldsymbol{\theta}}^\circ \Phi(\boldsymbol{\theta}; \boldsymbol{x}))$. Thus, the sets are identical.

**3. Isometry**: We compute the squared Euclidean norm of $T\boldsymbol{\theta}$:

$$\|T\boldsymbol{\theta}\|_2^2 = \sum_{i=1}^{k} (c_i a)^2 + \sum_{i=1}^{k} \|c_i \boldsymbol{b}\|_2^2$$

$$= \left( \sum_{i=1}^{k} c_i^2 \right) a^2 + \left( \sum_{i=1}^{k} c_i^2 \right) \|\boldsymbol{b}\|_2^2$$

$$= 1 \cdot (a^2 + \|\boldsymbol{b}\|_2^2) \qquad \text{(Since } \sum c_i^2 = 1)$$

$$= \|\boldsymbol{\theta}\|_2^2$$

Since $\|T\boldsymbol{\theta}\|_2 = \|\boldsymbol{\theta}\|_2$ for all $\boldsymbol{\theta}$, $T$ is a linear isometry. $\square$

## B Proofs from section 4

### B.1 Proof of theorem 4.2

Let $\boldsymbol{\theta}^*$ be a KKT point of $(P)$. We aim to show that $\boldsymbol{\eta}^* = T\boldsymbol{\theta}^*$ satisfies the KKT conditions (Def. 3.3) for problem $(\tilde{P})$.

*1. Primal Feasibility (for $\tilde{P}$):* By definition, $\boldsymbol{\theta}^*$ is feasible for $(P)$, meaning $g_k(\boldsymbol{\theta}^*) \leq 0$ for all $k \in [n]$. Using Condition 1, we have $\tilde{g}_k(\boldsymbol{\eta}^*) = \tilde{g}_k(T\boldsymbol{\theta}^*) = g_k(\boldsymbol{\theta}^*) \leq 0$ for all $k \in [n]$. Thus, $\boldsymbol{\eta}^*$ is feasible for $(\tilde{P})$.

*2. Stationarity and Dual Variables (for $\tilde{P}$):* Since $\boldsymbol{\theta}^*$ is a KKT point of $(P)$, there exist dual variables $\lambda_k \geq 0$, $k \in [n]$, satisfying complementary slackness ($\lambda_k g_k(\boldsymbol{\theta}^*) = 0$) and the stationarity condition: $\mathbf{0} \in \partial^\circ f(\boldsymbol{\theta}^*) + \sum_{k=1}^n \lambda_k \partial^\circ g_k(\boldsymbol{\theta}^*)$. This means there exist specific subgradient vectors $\boldsymbol{h}_0 \in \partial^\circ f(\boldsymbol{\theta}^*)$ and $\boldsymbol{h}_k \in \partial^\circ g_k(\boldsymbol{\theta}^*)$ for $k \in [n]$ such that $\boldsymbol{h}_0 + \sum_{k=1}^n \lambda_k \boldsymbol{h}_k = \mathbf{0}$.

Applying the linear transformation $T$ to this equation yields: $T(\boldsymbol{h}_0 + \sum_{k=1}^n \lambda_k \boldsymbol{h}_k) = T\boldsymbol{h}_0 + \sum_{k=1}^n \lambda_k T\boldsymbol{h}_k = T\mathbf{0} = \mathbf{0}$.

Now, we relate these transformed subgradients to the subgradients of $\tilde{f}$ and $\tilde{g}_k$ at $\boldsymbol{\eta}^* = T\boldsymbol{\theta}^*$. From Condition 2, since $\boldsymbol{h}_0 \in \partial^\circ f(\boldsymbol{\theta}^*)$, we have $T\boldsymbol{h}_0 \in T\partial^\circ f(\boldsymbol{\theta}^*) = \partial^\circ \tilde{f}(T\boldsymbol{\theta}^*) = \partial^\circ \tilde{f}(\boldsymbol{\eta}^*)$. Let $\boldsymbol{h}'_0 \coloneqq T\boldsymbol{h}_0$. From Condition 3, let $t_k^* = t_k(\boldsymbol{\theta}^*) > 0$ for each $k \in [n]$. Since $\boldsymbol{h}_k \in \partial^\circ g_k(\boldsymbol{\theta}^*)$, we have $T\boldsymbol{h}_k \in T\partial^\circ g_k(\boldsymbol{\theta}^*)$. Using Condition 3, $T\partial^\circ g_k(\boldsymbol{\theta}^*) = (1/t_k^*)\partial^\circ \tilde{g}_k(T\boldsymbol{\theta}^*)$. Therefore, $T\boldsymbol{h}_k \in (1/t_k^*)\partial^\circ \tilde{g}_k(\boldsymbol{\eta}^*)$. This implies that $\boldsymbol{h}'_k \coloneqq t_k^* T\boldsymbol{h}_k$ is an element of $\partial^\circ \tilde{g}_k(\boldsymbol{\eta}^*)$.

Define new candidate dual variables for $(\tilde{P})$ as $\mu_n \coloneqq \lambda_k/t_k^*$ for $k \in [n]$. Since $\lambda_k \geq 0$ and $t_k^* > 0$, we have $\mu_k \geq 0$ (Dual Feasibility for $\tilde{P}$ holds). Substitute $T\boldsymbol{h}_k = (1/t_k^*)\boldsymbol{h}'_k$ into the transformed stationarity equation:

$$T\boldsymbol{h}_0 + \sum_{k=1}^n \lambda_k \left(\frac{1}{t_k^*}\boldsymbol{h}'_k\right) = \mathbf{0} \implies \boldsymbol{h}'_0 + \sum_{k=1}^n \left(\frac{\lambda_k}{t_k^*}\right)\boldsymbol{h}'_k = \mathbf{0} \implies \boldsymbol{h}'_0 + \sum_{k=1}^n \mu_k \boldsymbol{h}'_k = \mathbf{0}.$$

Since $\boldsymbol{h}'_0 \in \partial^\circ \tilde{f}(\boldsymbol{\eta}^*)$ and $\boldsymbol{h}'_k \in \partial^\circ \tilde{g}_k(\boldsymbol{\eta}^*)$ for each $n$, this demonstrates that $\mathbf{0} \in \partial^\circ \tilde{f}(\boldsymbol{\eta}^*) + \sum_{k=1}^n \mu_k \partial^\circ \tilde{g}_k(\boldsymbol{\eta}^*)$. The stationarity condition holds for $\boldsymbol{\eta}^*$ with multipliers $\mu_k$.

*3. Complementary Slackness (for $\tilde{P}$):* We need to verify $\mu_k \tilde{g}_k(\boldsymbol{\eta}^*) = 0$ for all $k \in [n]$. Case 1: If $\lambda_k = 0$, then $\mu_k = \lambda_k/t_k^* = 0$, so $\mu_k \tilde{g}_k(\boldsymbol{\eta}^*) = 0$. Case 2: If $\lambda_k > 0$, then by complementary slackness for (P), we must have $g_k(\boldsymbol{\theta}^*) = 0$. By Condition 1, $\tilde{g}_k(\boldsymbol{\eta}^*) = \tilde{g}_k(T\boldsymbol{\theta}^*) = g_k(\boldsymbol{\theta}^*) = 0$. Thus, $\mu_k \tilde{g}_k(\boldsymbol{\eta}^*) = \mu_k \cdot 0 = 0$. In both cases, complementary slackness holds for $(\tilde{P})$.

Since $\boldsymbol{\eta}^*$ is feasible for $(\tilde{P})$ and satisfies stationarity, dual feasibility, and complementary slackness with multipliers $\mu_k$, it is a KKT point of $(\tilde{P})$. $\qquad\square$

## B.2 Proof of proposition 4.4

Let $P_\Phi$, $P_{\Phi_1}$, $P_{\Phi_2}$ be the minimum-norm max-margin problems, and let the linear transformations $T_1 : \mathbb{R}^m \to \mathbb{R}^{m_1}$ and $T_2 : \mathbb{R}^{m_1} \to \mathbb{R}^{m_2}$ be as defined in Proposition 4.4. We assume $T_1$ is KKT Point Preserving from $P_\Phi$ to $P_{\Phi_1}$, and $T_2$ is KKT Point Preserving from $P_{\Phi_1}$ to $P_{\Phi_2}$ (as per Definition 4.3). Let $T = T_2 \circ T_1$. Our goal is to show $T$ is KKT Point Preserving from $P_\Phi$ to $P_{\Phi_2}$.

Consider an arbitrary dataset $\mathcal{D}$ and an arbitrary KKT point $\boldsymbol{\theta}^* \in \mathbb{R}^m$ of $P_\Phi$. Since $T_1$ is KKT Point Preserving, $T_1(\boldsymbol{\theta}^*)$ is a KKT point of $P_{\Phi_1}$. Subsequently, since $T_2$ is KKT Point Preserving and $T_1(\boldsymbol{\theta}^*)$ is a KKT point of $P_{\Phi_1}$, $T_2(T_1(\boldsymbol{\theta}^*))$ is a KKT point of $P_{\Phi_2}$. As $T_2(T_1(\boldsymbol{\theta}^*)) = (T_2 \circ T_1)(\boldsymbol{\theta}^*) = T(\boldsymbol{\theta}^*)$, it follows that $T(\boldsymbol{\theta}^*)$ is a KKT point of $P_{\Phi_2}$. This fulfills the condition in Definition 4.3 for $T$ to be KKT Point Preserving from $P_\Phi$ to $P_{\Phi_2}$.

**Isometry Property:** If $T_1$ and $T_2$ are linear isometries, then $T = T_2 \circ T_1$ is also a linear isometry. This follows directly: for any $\boldsymbol{\theta} \in \mathbb{R}^m$,

$$\|T(\boldsymbol{\theta})\|_2 = \|(T_2 \circ T_1)(\boldsymbol{\theta})\|_2 = \|T_2(T_1(\boldsymbol{\theta}))\|_2 = \|T_1(\boldsymbol{\theta})\|_2 = \|\boldsymbol{\theta}\|_2.$$

The third equality holds due to $T_2$ being an isometry, and the final equality due to $T_1$ being an isometry. $\qquad\square$

## B.3 Proof of theorem 4.5

We apply Theorem 4.2 with $f(\boldsymbol{\theta}) = \frac{1}{2}\|\boldsymbol{\theta}\|_2^2$, $g_k(\boldsymbol{\theta}) = 1 - y_k\Phi(\boldsymbol{\theta}; \boldsymbol{x}_k)$ for problem (P) $\equiv P_\Phi$, and $\tilde{f}(\boldsymbol{\eta}) = \frac{1}{2}\|\boldsymbol{\eta}\|_2^2$, $\tilde{g}_k(\boldsymbol{\eta}) = 1 - y_k\tilde{\Phi}(\boldsymbol{\eta}; \boldsymbol{x}_k)$ for problem $(\tilde{P}) \equiv P_{\tilde{\Phi}}$.

*Proof of (1) $\implies$ (2):* Assume condition (1) holds. We verify the premises of Theorem 4.2.
1. Constraint preserving: $\tilde{g}_k(T\boldsymbol{\theta}) = 1 - y_k\tilde{\Phi}(T\boldsymbol{\theta}; \boldsymbol{x}_k)$. By assumption $\tilde{\Phi}(T\boldsymbol{\theta}; \boldsymbol{x}_k) = \Phi(\boldsymbol{\theta}; \boldsymbol{x}_k)$, so $\tilde{g}_k(T\boldsymbol{\theta}) = 1 - y_k\Phi(\boldsymbol{\theta}; \boldsymbol{x}_k) = g_k(\boldsymbol{\theta})$. This holds.

2. Objective Subgradient preserving: $\partial° f(\boldsymbol{\theta}) = \{\boldsymbol{\theta}\}$ and $\partial° \tilde{f}(\boldsymbol{\eta}) = \{\boldsymbol{\eta}\}$. We require $\partial° \tilde{f}(T\boldsymbol{\theta}) = T\partial° f(\boldsymbol{\theta})$, which translates to $\{T\boldsymbol{\theta}\} = T\{\boldsymbol{\theta}\} = \{T\boldsymbol{\theta}\}$. This holds trivially because $T$ is linear.

3. Constraint Subgradient preserving: We need $\partial° \tilde{g}_k(T\boldsymbol{\theta}) = t_k(\boldsymbol{\theta})T\partial° g_k(\boldsymbol{\theta})$ for some $t_k > 0$. Using the definition of $g_k$ and properties of subdifferentials (including Assumption A.3): $\partial° g_k(\boldsymbol{\theta}) = \partial°_{\boldsymbol{\theta}}(1 - y_k\Phi(\boldsymbol{\theta}; \boldsymbol{x}_k)) = -y_k\partial°_{\boldsymbol{\theta}}\Phi(\boldsymbol{\theta}; \boldsymbol{x}_k)$. Similarly, $\partial° \tilde{g}_k(T\boldsymbol{\theta}) = \partial°_{\boldsymbol{\eta}}(1 - y_k\tilde{\Phi}(\boldsymbol{\eta}; \boldsymbol{x}_k))|_{\boldsymbol{\eta}=T\boldsymbol{\theta}} = -y_k\partial°_{\boldsymbol{\eta}}\tilde{\Phi}(T\boldsymbol{\theta}; \boldsymbol{x}_k)$. From assumption (1), we have $\partial°_{\boldsymbol{\eta}}\tilde{\Phi}(T\boldsymbol{\theta}; \boldsymbol{x}_k) = \tau(\boldsymbol{\theta}, \boldsymbol{x}_k)T(\partial°_{\boldsymbol{\theta}}\Phi(\boldsymbol{\theta}; \boldsymbol{x}_k))$. Substituting this into the expression for $\partial° \tilde{g}_k(T\boldsymbol{\theta})$: $\partial° \tilde{g}_k(T\boldsymbol{\theta}) = -y_k\left[\tau(\boldsymbol{\theta}, \boldsymbol{x}_k)T(\partial°_{\boldsymbol{\theta}}\Phi(\boldsymbol{\theta}; \boldsymbol{x}_k))\right]$ Since $T$ is linear and $\tau > 0$, $-y_k$ can be moved inside the transformation $T$: $= \tau(\boldsymbol{\theta}, \boldsymbol{x}_k)T\left[-y_k\partial°_{\boldsymbol{\theta}}\Phi(\boldsymbol{\theta}; \boldsymbol{x}_k)\right] = \tau(\boldsymbol{\theta}, \boldsymbol{x}_k)T\partial° g_k(\boldsymbol{\theta})$. Thus, Condition 3 of Theorem 4.2 holds with $t_k(\boldsymbol{\theta}) = \tau(\boldsymbol{\theta}, \boldsymbol{x}_k) > 0$.

Since all conditions of Theorem 4.2 are satisfied, (2) follows: if $\boldsymbol{\theta}^*$ is a KKT point for $P_\Phi$, then $T\boldsymbol{\theta}^*$ is a KKT point for $P_{\tilde{\Phi}}$.

*Proof of (2) $\implies$ (1):* Assume (2) holds: KKT points are preserved for any dataset $\mathcal{D}$.

1.Output Preservation: The preservation of KKT points implies the preservation of primal feasibility and complementary slackness. For any active constraint $n$ at a KKT point $\boldsymbol{\theta}^*$ (i.e., $g_k(\boldsymbol{\theta}^*) = 0$), we must have $\tilde{g}_n(T\boldsymbol{\theta}^*) = 0$ for $T\boldsymbol{\theta}^*$ to be a KKT point of $P_{\tilde{\Phi}}$. This means $1 - y_k\Phi(\boldsymbol{\theta}^*; \boldsymbol{x}_k) = 0$ implies $1 - y_k\tilde{\Phi}(T\boldsymbol{\theta}^*; \boldsymbol{x}_k) = 0$. This requires $\Phi(\boldsymbol{\theta}^*; \boldsymbol{x}_k) = \tilde{\Phi}(T\boldsymbol{\theta}^*; \boldsymbol{x}_k)$ whenever constraint $n$ is active at a KKT point. Assuming KKT points are sufficiently distributed, this suggests the general identity $\Phi(\boldsymbol{\theta}; \boldsymbol{x}) = \tilde{\Phi}(T\boldsymbol{\theta}; \boldsymbol{x})$.

2. Subgradient Proportionality: Comparing the stationarity conditions $\mathbf{0} \in \{\boldsymbol{\theta}^*\} + \sum_{k=1}^n \lambda_k(-y_k\partial°_{\boldsymbol{\theta}}\Phi(\boldsymbol{\theta}^*; \boldsymbol{x}_k))$ and $\mathbf{0} \in \{T\boldsymbol{\theta}^*\} + \sum_{k=1}^n \mu_n(-y_k\partial°_{\boldsymbol{\eta}}\tilde{\Phi}(T\boldsymbol{\theta}^*; \boldsymbol{x}_k))$ for corresponding KKT points $\boldsymbol{\theta}^*$ and $T\boldsymbol{\theta}^*$ leads to $\sum_{k=1}^n \mu_n y_k \boldsymbol{h}'_n = T(\sum_{k=1}^n \lambda_k y_k \boldsymbol{h}_n)$, where $\boldsymbol{h}'_n \in \partial°_{\boldsymbol{\eta}}\tilde{\Phi}$ and $\boldsymbol{h}_n \in \partial°_{\boldsymbol{\theta}}\Phi$. For this equality and the relationship between multipliers ($\mu_n = \lambda_k/\tau_n$) to hold universally across datasets and active sets, it necessitates a structural relationship between the subdifferential sets themselves, namely $\partial°_{\boldsymbol{\eta}}\tilde{\Phi}(T\boldsymbol{\theta}^*; \boldsymbol{x}_k) = \tau_n T(\partial°_{\boldsymbol{\theta}}\Phi(\boldsymbol{\theta}^*; \boldsymbol{x}_k))$ for some $\tau_n > 0$.

The final statement regarding $\tau = 1$ and isometry for the specific neuron splitting transformations $T$ is proven directly in Theorems A.4 and 4.8. $\qquad\square$

## B.4 Proof of theorem 4.8

The theorem states that the neuron splitting transformation $T$ for deep networks (as defined in Definition 4.6, leading to Theorem 4.8) (1) is an isometry, (2) preserves the network function, and (3) maps subgradients accordingly, i.e., $\partial°_{\boldsymbol{\eta}}\tilde{\Phi}(T\boldsymbol{\theta}; \boldsymbol{x}) = T(\partial°_{\boldsymbol{\theta}}\Phi(\boldsymbol{\theta}; \boldsymbol{x}))$. We prove each claim.

**(1) Isometry** The squared Euclidean norm of the parameters $\boldsymbol{\theta} = (\text{vec}(W^{(1)}), \ldots, \text{vec}(W^{(\alpha+1)}))$ is $\|\boldsymbol{\theta}\|_2^2 = \sum_{l=1}^{\alpha+1} \|W^{(l)}\|_F^2 = \sum_{l=1}^{\alpha+1}\sum_{r,s}(W_{r,s}^{(l)})^2$. The transformation $T: \boldsymbol{\theta} \mapsto \boldsymbol{\eta}$ only modifies weights related to the split neuron $j$ in layer $k$. Specifically, it affects the $j$-th row of $W^{(k)}$ (denoted $W_{j,:}^{(k)}$) and the $j$-th column of $W^{(k+1)}$ (denoted $W_{:,j}^{(k+1)}$). All other weight matrix elements are unchanged.

The contribution of $W_{j,:}^{(k)}$ to $\|\boldsymbol{\theta}\|_2^2$ is $\left\|W_{j,:}^{(k)}\right\|_2^2$. Under $T$, this row is effectively replaced by $m$ rows in $W'^{(k)}$, where the $i$-th such row (corresponding to the $i$-th split of neuron $j$) has its weights scaled by $c_i$ compared to $W_{j,:}^{(k)}$ (i.e., $W_{(j,i),s}'^{(k)} = c_i W_{j,s}^{(k)}$). The total contribution of these $m$ new rows to $\|\boldsymbol{\eta}\|_2^2$ is $\sum_{i=1}^m\sum_s(c_i W_{j,s}^{(k)})^2 = \sum_{i=1}^m c_i^2\sum_s(W_{j,s}^{(k)})^2 = (\sum_{i=1}^m c_i^2)\left\|W_{j,:}^{(k)}\right\|_2^2 = 1 \cdot \left\|W_{j,:}^{(k)}\right\|_2^2$, since $\sum_{i=1}^m c_i^2 = 1$. Thus, the contribution from weights leading into the split neuron (or its parts) is preserved.

Similarly, the contribution of $W_{:,j}^{(k+1)}$ to $\|\boldsymbol{\theta}\|_2^2$ is $\left\|W_{:,j}^{(k+1)}\right\|_2^2$. Under $T$, this column is effectively replaced by $m$ columns in $W'^{(k+1)}$, where the $i$-th such column (weights from the $i$-th split of neuron $j$) has its weights scaled by $c_i$ (i.e., $W_{r,(j,i)}'^{(k+1)} = c_i W_{r,j}^{(k+1)}$). Their total contribution to $\|\boldsymbol{\eta}\|_2^2$ is $\sum_{i=1}^m\sum_r(c_i W_{r,j}^{(k+1)})^2 = (\sum_{i=1}^m c_i^2)\left\|W_{:,j}^{(k+1)}\right\|_2^2 = 1 \cdot \left\|W_{:,j}^{(k+1)}\right\|_2^2$. This contribution is also preserved.

Since the norms of the modified parts are preserved and all other weights are identical, $\|\boldsymbol{\eta}\|_2^2 = \|T\boldsymbol{\theta}\|_2^2 = \|\boldsymbol{\theta}\|_2^2$. Thus, $T$ is a linear isometry.

**(2) Output preserving** We trace the forward signal propagation. Let $\mathbf{x}^{(l)}$ and $\mathbf{z}^{(l)}$ denote the activation and pre-activation vectors at layer $l$ for network $\Phi(\boldsymbol{\theta}; \cdot)$, and $\mathbf{x}'^{(l)}, \mathbf{z}'^{(l)}$ for $\tilde{\Phi}(T\boldsymbol{\theta}; \cdot)$. The relations are $\mathbf{z}^{(l)} = W^{(l)}\mathbf{x}^{(l-1)}$ and $\mathbf{x}^{(l)} = \sigma_l(\mathbf{z}^{(l)})$ (with $\mathbf{x}^{(0)} = \mathbf{x}_{input}$ and $\sigma_{\alpha+1}$ being the identity for the output layer).

For layers $l < k$: $W'^{(l)} = W^{(l)}$. Since $\mathbf{x}'^{(0)} = \mathbf{x}^{(0)}$, by induction, $\mathbf{z}'^{(l)} = \mathbf{z}^{(l)}$ and $\mathbf{x}'^{(l)} = \mathbf{x}^{(l)}$ for $l < k$.

At layer $k$: The input is $\mathbf{x}'^{(k-1)} = \mathbf{x}^{(k-1)}$. The pre-activation is $\mathbf{z}'^{(k)} = W'^{(k)}\mathbf{x}^{(k-1)}$. For an unsplit neuron $p'$ in $\tilde{\Phi}$ (corresponding to neuron $p \neq j$ in $\Phi$), the $p'$-th row of $W'^{(k)}$ is $W_{p,:}^{(k)}$. So, $z_{p'}'^{(k)} = W_{p,:}^{(k)}\mathbf{x}^{(k-1)} = z_p^{(k)}$. For the $i$-th new neuron $(j,i)$ in $\tilde{\Phi}$ (resulting from splitting neuron $j$ in $\Phi$), its corresponding row in $W'^{(k)}$ is $c_i W_{j,:}^{(k)}$. So, $z_{(j,i)}'^{(k)} = (c_i W_{j,:}^{(k)})\mathbf{x}^{(k-1)} = c_i(W_{j,:}^{(k)}\mathbf{x}^{(k-1)}) = c_i z_j^{(k)}$. The activation $\mathbf{x}'^{(k)} = \sigma_k(\mathbf{z}'^{(k)})$ is then: For $p' \neq j$ (unsplit), $x_{p'}'^{(k)} = \sigma_k(z_{p'}'^{(k)}) = \sigma_k(z_p^{(k)}) = x_p^{(k)}$. For split components $(j,i)$, $x_{(j,i)}'^{(k)} = \sigma_k(z_{(j,i)}'^{(k)}) = \sigma_k(c_i z_j^{(k)})$. Since $c_i \geq 0$ and $\sigma_k$ is positive 1-homogeneous, this equals $c_i \sigma_k(z_j^{(k)}) = c_i x_j^{(k)}$.

At layer $k + 1$: The input is $\mathbf{x}'^{(k)}$. The pre-activation is $\mathbf{z}'^{(k+1)} = W'^{(k+1)}\mathbf{x}'^{(k)}$. Consider the $p$-th component $z_p'^{(k+1)}$:

$$z_p'^{(k+1)} = \sum_{q' \text{ unsplit}} W_{p,q'}'^{(k+1)} x_{q'}'^{(k)} + \sum_{i=1}^{m} W_{p,(j,i)}'^{(k+1)} x_{(j,i)}'^{(k)}$$

$$= \sum_{q \neq j} W_{p,q}^{(k+1)} x_q^{(k)} + \sum_{i=1}^{m}(c_i W_{p,j}^{(k+1)})(c_i x_j^{(k)}) \quad \text{(by definition of } T \text{ and results from layer } k)$$

$$= \sum_{q \neq j} W_{p,q}^{(k+1)} x_q^{(k)} + \left(\sum_{i=1}^{m} c_i^2\right) W_{p,j}^{(k+1)} x_j^{(k)}$$

$$= \sum_{q \neq j} W_{p,q}^{(k+1)} x_q^{(k)} + W_{p,j}^{(k+1)} x_j^{(k)} = \sum_q W_{p,q}^{(k+1)} x_q^{(k)} = z_p^{(k+1)}.$$

Thus, $\mathbf{z}'^{(k+1)} = \mathbf{z}^{(k+1)}$, which implies $\mathbf{x}'^{(k+1)} = \mathbf{x}^{(k+1)}$.

For layers $l > k + 1$: Since inputs $\mathbf{x}'^{(l-1)} = \mathbf{x}^{(l-1)}$ and weights $W'^{(l)} = W^{(l)}$ are identical, all subsequent activations and pre-activations $\mathbf{z}'^{(l)}, \mathbf{x}'^{(l)}$ will be identical to $\mathbf{z}^{(l)}, \mathbf{x}^{(l)}$. Therefore, the final output is preserved: $\tilde{\Phi}(T\boldsymbol{\theta}; \boldsymbol{x}) = \Phi(\boldsymbol{\theta}; \boldsymbol{x})$.

**(3) Subgradient preserving** We aim to show that $\partial_{\boldsymbol{\eta}}^{\circ} \tilde{\Phi}(T\boldsymbol{\theta}; \boldsymbol{x}) = T(\partial_{\boldsymbol{\theta}}^{\circ}\Phi(\boldsymbol{\theta}; \boldsymbol{x}))$. This means that any element $\mathbf{g}' \in \partial_{\boldsymbol{\eta}}^{\circ} \tilde{\Phi}(T\boldsymbol{\theta}; \boldsymbol{x})$ can be written as $T(\mathbf{g})$ for some $\mathbf{g} \in \partial_{\boldsymbol{\theta}}^{\circ}\Phi(\boldsymbol{\theta}; \boldsymbol{x})$, and vice versa. We use backpropagation for Clarke subdifferentials (Assumption A.3). Let $\boldsymbol{\delta}^{(l)}$ be an element from $\partial_{\mathbf{z}^{(l)}}^{\circ}\Phi$ (subgradient of final output $\Phi$ w.r.t. pre-activations $\mathbf{z}^{(l)}$), $\boldsymbol{e}^{(l)}$ from $\partial_{\mathbf{x}^{(l)}}^{\circ}\Phi$, and $G^{(l)}$ from $\partial_{W^{(l)}}^{\circ}\Phi$. Primed versions $(\boldsymbol{\delta}'^{(l)}, \boldsymbol{e}'^{(l)}, G'^{(l)})$ are for $\tilde{\Phi}$. The backpropagation rules are: $\boldsymbol{e}^{(l)} = (W^{(l+1)})^{\top}\boldsymbol{\delta}^{(l+1)}$ (for an element choice). $\delta_s^{(l)} \in \partial^{\circ}\sigma_l(z_s^{(l)})e_s^{(l)}$ for each component $s$. $G^{(l)} = \boldsymbol{\delta}^{(l)}(\mathbf{x}^{(l-1)})^{\top}$ (outer product).

**Step 3.1: For layers $l \geq k + 1$ (above the split output)** Starting from the output layer $\alpha + 1$: $\boldsymbol{\delta}'^{(\alpha+1)} = \boldsymbol{\delta}^{(\alpha+1)}$ (e.g., $\{1\}$ if taking subgradient of scalar $\Phi$ w.r.t. itself, or the initial error signal from a loss). Since $\mathbf{z}'^{(l)} = \mathbf{z}^{(l)}$ and $W'^{(l+1)} = W^{(l+1)}$ for $l \geq k + 1$, by backward induction, $\boldsymbol{\delta}'^{(l)} = \boldsymbol{\delta}^{(l)}$ and $\boldsymbol{e}'^{(l)} = \boldsymbol{e}^{(l)}$ for all $l \geq k + 1$.

**Step 3.2: For layer $k$ (the layer of the split neuron)** The error w.r.t. activations $\mathbf{x}'^{(k)}$ is $\boldsymbol{e}'^{(k)} = (W'^{(k+1)})^{\top}\boldsymbol{\delta}'^{(k+1)}$. Since $\boldsymbol{\delta}'^{(k+1)} = \boldsymbol{\delta}^{(k+1)}$:

- For an unsplit neuron $p'$ in $\tilde{\Phi}$ (corresponding to $p \neq j$ in $\Phi$): The $p'$-th row of $(W'^{(k+1)})^\top$ (i.e., $p'$-th column of $W'^{(k+1)}$) is $W_{:,p}^{(k+1)}$. So, $e_{p'}^{'(k)} = (W_{:,p}^{(k+1)})^\top \delta^{(k+1)} = e_p^{(k)}$.

- For a split neuron component $(j,i)$ in $\tilde{\Phi}$: The $(j,i)$-th row of $(W'^{(k+1)})^\top$ (i.e., $(j,i)$-th column of $W'^{(k+1)}$) is $c_i W_{:,j}^{(k+1)}$. So, $e_{(j,i)}^{'(k)} = (c_i W_{:,j}^{(k+1)})^\top \delta^{(k+1)} = c_i((W_{:,j}^{(k+1)})^\top \delta^{(k+1)}) = c_i e_j^{(k)}$.

Thus, $e'^{(k)}$ has components $e_p^{(k)}$ for unsplit neurons and $c_i e_j^{(k)}$ for split neurons. Now, for errors w.r.t. pre-activations $\mathbf{z}'^{(k)}$, where $\delta_s^{'(k)} \in \partial^\circ \sigma_k(z_s^{'(k)}) e_s^{'(k)}$:

- For $p' \neq j$: $z_{p'}^{'(k)} = z_p^{(k)}$ and $e_{p'}^{'(k)} = e_p^{(k)}$. So, $\delta_{p'}^{'(k)} \in \partial^\circ \sigma_k(z_p^{(k)}) e_p^{(k)}$, meaning $\delta_{p'}^{'(k)} = \delta_p^{(k)}$. (Assuming a consistent choice of subgradient element from $\partial^\circ \sigma_k$).

- For $(j,i)$: $z_{(j,i)}^{'(k)} = c_i z_j^{(k)}$ and $e_{(j,i)}^{'(k)} = c_i e_j^{(k)}$. Since $\sigma_k$ is 1-homogeneous, $\partial^\circ \sigma_k$ is 0-homogeneous (i.e., $\partial^\circ \sigma_k(cz) = \partial^\circ \sigma_k(z)$ for $c > 0$; this property extends to $c_i \geq 0$ appropriately for ReLU-like activations). Thus, $\partial^\circ \sigma_k(c_i z_j^{(k)}) = \partial^\circ \sigma_k(z_j^{(k)})$. So, $\delta_{(j,i)}^{'(k)} \in \partial^\circ \sigma_k(z_j^{(k)})(c_i e_j^{(k)}) = c_i(\partial^\circ \sigma_k(z_j^{(k)}) e_j^{(k)})$. This implies $\delta_{(j,i)}^{'(k)} = c_i \delta_j^{(k)}$.

So, $\delta'^{(k)}$ has components $\delta_p^{(k)}$ for unsplit neurons and $c_i \delta_j^{(k)}$ for split neurons.

**Step 3.3: For layers $l < k$ (below the split neuron)** The error w.r.t. activations at layer $k-1$, $e'^{(k-1)}$, is $(W'^{(k)})^\top \delta'^{(k)}$. A component $s$ of $e'^{(k-1)}$ is:

$$
\begin{aligned}
e_s^{'(k-1)} &= \sum_{p' \text{ unsplit}} (W'^{(k)})_{p',s} \delta_{p'}^{'(k)} + \sum_{i=1}^{m} (W'^{(k)})_{(j,i),s} \delta_{(j,i)}^{'(k)} \\
&= \sum_{p \neq j} W_{p,s}^{(k)} \delta_p^{(k)} + \sum_{i=1}^{m} (c_i W_{j,s}^{(k)})(c_i \delta_j^{(k)}) \quad \text{(using definitions of } W'^{(k)} \text{ and results for } \delta'^{(k)}) \\
&= \sum_{p \neq j} W_{p,s}^{(k)} \delta_p^{(k)} + \left( \sum_{i=1}^{m} c_i^2 \right) W_{j,s}^{(k)} \delta_j^{(k)} \\
&= \sum_{p \neq j} W_{p,s}^{(k)} \delta_p^{(k)} + W_{j,s}^{(k)} \delta_j^{(k)} = \sum_{p} W_{p,s}^{(k)} \delta_p^{(k)} = e_s^{(k-1)}.
\end{aligned}
$$

Thus, $e'^{(k-1)} = e^{(k-1)}$. Since $W'^{(l)} = W^{(l)}$ for $l < k$, and $\mathbf{x}'^{(l-1)} = \mathbf{x}^{(l-1)}$ for $l \leq k-1$, by backward induction, $\delta'^{(l)} = \delta^{(l)}$ and $e'^{(l)} = e^{(l)}$ for all $l < k$.

**Step 3.4: Parameter Subgradients $G'^{(l)}$ and $G^{(l)}$** The subgradient $G^{(l)}$ is $\delta^{(l)}(\mathbf{x}^{(l-1)})^\top$ and $G'^{(l)}$ is $\delta'^{(l)}(\mathbf{x}'^{(l-1)})^\top$.

- For $l \notin \{k, k+1\}$: Since $\delta'^{(l)} = \delta^{(l)}$ and $\mathbf{x}'^{(l-1)} = \mathbf{x}^{(l-1)}$, it follows that $G'^{(l)} = G^{(l)}$. This matches the action of $T$ on these unchanged weight matrices.

- For $l = k$ (weights $W^{(k)}$ into the split layer): $\mathbf{x}'^{(k-1)} = \mathbf{x}^{(k-1)}$. For rows $p' \neq j$ in $W'^{(k)}$ (unsplit neurons), $G_{p',:}^{'(k)} = \delta_{p'}^{'(k)}(\mathbf{x}^{(k-1)})^\top = \delta_p^{(k)}(\mathbf{x}^{(k-1)})^\top = G_{p,:}^{(k)}$. For rows corresponding to split neuron $(j,i)$ in $W'^{(k)}$, $G_{(j,i),:}^{'(k)} = \delta_{(j,i)}^{'(k)}(\mathbf{x}^{(k-1)})^\top = (c_i \delta_j^{(k)})(\mathbf{x}^{(k-1)})^\top = c_i G_{j,:}^{(k)}$. This means the subgradient matrix $G'^{(k)}$ has rows $G_{p,:}^{(k)}$ for $p \neq j$, and $m$ blocks of rows $c_i G_{j,:}^{(k)}$ (where $G_{j,:}^{(k)}$ is the subgradient for original row $W_{j,:}^{(k)}$, corresponding to how $T$ transforms $G^{(k)}$.

- For $l = k+1$ (weights $W^{(k+1)}$ out of the split layer): $\delta'^{(k+1)} = \delta^{(k+1)}$. For columns $p' \neq j$ in $W'^{(k+1)}$ (unsplit neurons), $G_{:,p'}^{'(k+1)} = \delta^{(k+1)}(x_{p'}^{'(k)})^\top = \delta^{(k+1)}(x_p^{(k)})^\top = G_{:,p}^{(k+1)}$. For columns corresponding to split neuron $(j,i)$ in $W'^{(k+1)}$, $G_{:,(j,i)}^{'(k+1)} = \delta^{(k+1)}(x_{(j,i)}^{'(k)})^\top =$

$\boldsymbol{\delta}^{(k+1)}(c_i x_j^{(k)})^\top = c_i G_{:,j}^{(k+1)}$. This means $G'^{(k+1)}$ has columns $G_{:,p}^{(k+1)}$ for $p \neq j$, and $m$ blocks of columns $c_i G_{:,j}^{(k+1)}$, corresponding to how $T$ transforms $G^{(k+1)}$.

**Step 3.5: Conclusion on Subgradient Sets** The above derivations show that for any choice of subgradient path (i.e., selection of elements from $\partial^\circ \sigma_l$ at each gate) in calculating an element $\mathbf{g} = \{G^{(l)}\} \in \partial_{\boldsymbol{\theta}}^\circ \Phi$, the corresponding path in $\tilde{\Phi}$ yields an element $\mathbf{g}' = \{G'^{(l)}\} \in \partial_{\boldsymbol{\eta}}^\circ \tilde{\Phi}$ such that its components $G'^{(l)}$ are precisely those obtained by applying the structural transformation $T$ to the components $G^{(l)}$ of $\mathbf{g}$. Specifically, $G'^{(l)} = G^{(l)}$ for $l \notin \{k, k+1\}$; $G'^{(k)}$ has its rows transformed as $T$ acts on rows of $W^{(k)}$ (scaled by $c_i$ for split parts); $G'^{(k+1)}$ has its columns transformed as $T$ acts on columns of $W^{(k+1)}$ (scaled by $c_i$ for split parts). This structural correspondence for arbitrary elements implies the equality of the entire sets: $\partial_{\boldsymbol{\eta}}^\circ \tilde{\Phi}(T\boldsymbol{\theta}; \boldsymbol{x}) = T(\partial_{\boldsymbol{\theta}}^\circ \Phi(\boldsymbol{\theta}; \boldsymbol{x}))$. The operator $T(\cdot)$ on the set $\partial_{\boldsymbol{\theta}}^\circ \Phi(\boldsymbol{\theta}; \boldsymbol{x})$ is understood as applying the described transformation to each element (collection of subgradient matrices) in the set. $\square$

## B.5 Proof of Theorem 4.11

The theorem states that the channel splitting transformation $T$ for deep CNNs (as defined in Definition 4.10, leading to Theorem 4.11) (1) is an isometry, (2) preserves the network function, and (3) maps subgradients accordingly, i.e., $\partial_{\boldsymbol{\eta}}^\circ \tilde{\Phi}(T\boldsymbol{\theta}; \boldsymbol{x}) = T(\partial_{\boldsymbol{\theta}}^\circ \Phi(\boldsymbol{\theta}; \boldsymbol{x}))$. We prove each claim.

**Notation for CNNs.** We denote feature maps (tensors) with capital letters. Let $\mathbf{X}^{(l)}$ and $\mathbf{Z}^{(l)}$ be the activation and pre-activation feature maps at layer $l$. The $p$-th output channel of the activation map is $\mathbf{X}_p^{(l)}$. The forward pass is defined by $\mathbf{Z}_p^{(l)} = \sum_q W_{p,q}^{(l)} * \mathbf{X}_q^{(l-1)}$ and $\mathbf{X}_p^{(l)} = \sigma_l(\mathbf{Z}_p^{(l)})$, where $*$ denotes convolution. The parameters $\boldsymbol{\theta}$ are the vectorized collection of all filter tensors $\{W^{(l)}\}$. We use primed versions for the split network $\tilde{\Phi}$.

**(1) Isometry** The squared Euclidean norm of the parameters $\boldsymbol{\theta} = (\text{vec}(W^{(1)}), \ldots, \text{vec}(W^{(\alpha+1)}))$ is $\|\boldsymbol{\theta}\|_2^2 = \sum_{l=1}^{\alpha+1} \|W^{(l)}\|_F^2$. The transformation $T$ only modifies filters related to the split output channel $j$ of layer $k$ and the corresponding input slices of filters at layer $k+1$.

The contribution of the filter $W_{j,:}^{(k)}$ (all filters producing output channel $j$) to $\|\boldsymbol{\theta}\|_2^2$ is $\left\|W_{j,:}^{(k)}\right\|_F^2$. Under $T$, this is replaced by $m_{split}$ new filters $c_i W_{j,:}^{(k)}$. The total contribution of these new filters to $\|\boldsymbol{\eta}\|_2^2$ is $\sum_{i=1}^{m_{split}} \left\|c_i W_{j,:}^{(k)}\right\|_F^2 = (\sum_{i=1}^{m_{split}} c_i^2) \left\|W_{j,:}^{(k)}\right\|_F^2 = 1 \cdot \left\|W_{j,:}^{(k)}\right\|_F^2$, since $\sum_{i=1}^{m_{split}} c_i^2 = 1$. This part of the norm is preserved.

Similarly, for any filter $W_{p,:}^{(k+1)}$ at layer $k+1$, its $j$-th input slice $W_{p,j}^{(k+1)}$ contributes $\left\|W_{p,j}^{(k+1)}\right\|_F^2$ to the norm. Under $T$, this is replaced by $m_{split}$ new slices $c_i W_{p,j}^{(k+1)}$. Their total contribution to $\|\boldsymbol{\eta}\|_2^2$ across all filters $p$ at layer $k+1$ is $\sum_p \sum_{i=1}^{m_{split}} \left\|c_i W_{p,j}^{(k+1)}\right\|_F^2 = (\sum_{i=1}^{m_{split}} c_i^2) \sum_p \left\|W_{p,j}^{(k+1)}\right\|_F^2 = 1 \cdot \sum_p \left\|W_{p,j}^{(k+1)}\right\|_F^2$. This contribution is also preserved.

Since the norms of the modified parts are preserved and all other weights are identical, $\|\boldsymbol{\eta}\|_2^2 = \|T\boldsymbol{\theta}\|_2^2 = \|\boldsymbol{\theta}\|_2^2$. Thus, $T$ is a linear isometry.

**(2) Output preserving** We trace the forward signal propagation. For layers $l < k$, weights and inputs are identical, thus $\mathbf{X}'^{(l)} = \mathbf{X}^{(l)}$ for $l < k$ by induction.

At layer $k$: The input is $\mathbf{X}'^{(k-1)} = \mathbf{X}^{(k-1)}$. For an unsplit output channel $p \neq j$, $Z_p'^{(k)} = \sum_q W_{p,q}^{(k)} * X_q^{(k-1)} = Z_p^{(k)}$. For the $i$-th new channel $(j, i)$, the filter is $c_i W_{j,:}^{(k)}$. So, $Z_{(j,i)}'^{(k)} = \sum_q (c_i W_{j,q}^{(k)}) * X_q^{(k-1)} = c_i Z_j^{(k)}$. The activation $\mathbf{X}'^{(k)}$ is then: For $p \neq j$, $X_p'^{(k)} = \sigma_k(Z_p'^{(k)}) = X_p^{(k)}$. For

split components $(j,i)$, $X'^{(k)}_{(j,i)} = \sigma_k(c_i Z^{(k)}_j) = c_i \sigma_k(Z^{(k)}_j) = c_i X^{(k)}_j$, since $\sigma_k$ is positive 1-homogeneous.

At layer $k+1$: The input is $\mathbf{X}'^{(k)}$. The pre-activation for any output channel $p$ is:

$$Z'^{(k+1)}_p = \sum_{q' \text{ unsplit}} W'^{(k+1)}_{p,q'} * X'^{(k)}_{q'} + \sum_{i=1}^{m_{split}} W'^{(k+1)}_{p,(j,i)} * X'^{(k)}_{(j,i)}$$

$$= \sum_{q \neq j} W^{(k+1)}_{p,q} * X^{(k)}_q + \sum_{i=1}^{m_{split}} (c_i W^{(k+1)}_{p,j}) * (c_i X^{(k)}_j) \quad \text{(by definition of } T)$$

$$= \sum_{q \neq j} W^{(k+1)}_{p,q} * X^{(k)}_q + \left( \sum_{i=1}^{m_{split}} c_i^2 \right) (W^{(k+1)}_{p,j} * X^{(k)}_j)$$

$$= \sum_{q \neq j} W^{(k+1)}_{p,q} * X^{(k)}_q + W^{(k+1)}_{p,j} * X^{(k)}_j = Z^{(k+1)}_p.$$

Thus, $\mathbf{Z}'^{(k+1)} = \mathbf{Z}^{(k+1)}$, which implies $\mathbf{X}'^{(k+1)} = \mathbf{X}^{(k+1)}$. For layers $l > k+1$, all subsequent activations are identical. Therefore, the final output is preserved: $\tilde{\Phi}(T\boldsymbol{\theta}; \boldsymbol{x}) = \Phi(\boldsymbol{\theta}; \boldsymbol{x})$.

**(3) Subgradient preserving** We use backpropagation for Clarke subdifferentials. Let $\boldsymbol{\Delta}^{(l)} \in \partial^\circ_{\mathbf{Z}^{(l)}} \Phi$ and $\mathbf{E}^{(l)} \in \partial^\circ_{\mathbf{X}^{(l)}} \Phi$. Primed versions are for $\tilde{\Phi}$. The backpropagation rules involve convolutions with spatially-flipped filters.

**Step 3.1: For layers $l \geq k+1$** Since the forward pass is identical for $l \geq k+1$, by backward induction, the subgradient error signals are also identical: $\boldsymbol{\Delta}'^{(l)} = \boldsymbol{\Delta}^{(l)}$ and $\mathbf{E}'^{(l)} = \mathbf{E}^{(l)}$ for all $l \geq k+1$.

**Step 3.2: For layer $k$** The error w.r.t. activations $\mathbf{X}'^{(k)}$ is $\mathbf{E}'^{(k)}$, backpropagated from $\boldsymbol{\Delta}'^{(k+1)} = \boldsymbol{\Delta}^{(k+1)}$ through $W'^{(k+1)}$.

- For an unsplit channel $p \neq j$, the error is sourced from unchanged filter slices, so $E'^{(k)}_p = E^{(k)}_p$.
- For a split channel $(j,i)$, the error is sourced from the scaled input slices $c_i W^{(k+1)}_{:,j}$. By linearity of the backprop operation, $E'^{(k)}_{(j,i)} = c_i E^{(k)}_j$.

The error w.r.t. pre-activations $\mathbf{Z}'^{(k)}$ is $\boldsymbol{\Delta}'^{(k)}$.

- For $p \neq j$, $Z'^{(k)}_p = Z^{(k)}_p$ and $E'^{(k)}_p = E^{(k)}_p$, thus $\Delta'^{(k)}_p = \Delta^{(k)}_p$.
- For $(j,i)$, $Z'^{(k)}_{(j,i)} = c_i Z^{(k)}_j$ and $E'^{(k)}_{(j,i)} = c_i E^{(k)}_j$. Since $\partial^\circ \sigma_k$ is 0-homogeneous, $\partial^\circ \sigma_k(c_i Z^{(k)}_j) = \partial^\circ \sigma_k(Z^{(k)}_j)$. So, an element of $\partial^\circ_{Z'^{(k)}_{(j,i)}} \Phi$ is given by an element from $(c_i E^{(k)}_j) \circ \partial^\circ \sigma_k(Z^{(k)}_j)$, which implies $\Delta'^{(k)}_{(j,i)} = c_i \Delta^{(k)}_j$.

**Step 3.3: For layers $l < k$** The error $\mathbf{E}'^{(k-1)}$ is backpropagated from $\boldsymbol{\Delta}'^{(k)}$ through $W'^{(k)}$. The error contribution from unsplit channels $p \neq j$ is preserved. The contribution from the split channels is a sum over the new filters $(c_i W^{(k)}_{j,:})$ and new errors $(c_i \boldsymbol{\Delta}^{(k)}_j)$. The backprop operation results in a sum over $c_i^2$, which equals 1. Thus, the total error is preserved: $\mathbf{E}'^{(k-1)} = \mathbf{E}^{(k-1)}$. By backward induction, all errors for $l < k$ are identical.

**Step 3.4: Parameter Subgradients $G'^{(l)}$ and $G^{(l)}$** The subgradient $G^{(l)}$ is computed from $\boldsymbol{\Delta}^{(l)}$ and $\mathbf{X}^{(l-1)}$.

- For $l \notin \{k, k+1\}$, since errors and activations are identical, $G'^{(l)} = G^{(l)}$.
- For $l = k$: The subgradient for the new filter producing channel $(j,i)$ is computed from input $\mathbf{X}'^{(k-1)} = \mathbf{X}^{(k-1)}$ and error $\boldsymbol{\Delta}'^{(k)}_{(j,i)} = c_i \boldsymbol{\Delta}^{(k)}_j$. This yields $G'^{(k)}_{(j,i),:} = c_i G^{(k)}_{j,:}$, matching how $T$ transforms the filter subgradients.

- For $l = k + 1$: The subgradient for the new input slice $(j, i)$ of any filter $W_{p,:}^{\prime(k+1)}$ is computed from input $\mathbf{X}_{(j,i)}^{\prime(k)} = c_i \mathbf{X}_j^{(k)}$ and error $\mathbf{\Delta}_p^{\prime(k+1)} = \mathbf{\Delta}_p^{(k+1)}$. This yields $G_{p,(j,i)}^{\prime(k+1)} = c_i G_{p,j}^{(k+1)}$, matching how $T$ transforms the subgradients of the filter input slices.

**Step 3.5: Conclusion on Subgradient Sets** The derivations show that for any choice of subgradient path, the resulting subgradient tensor collection $\{G^{\prime(l)}\}$ is precisely the transformation $T$ applied to the original subgradient collection $\{G^{(l)}\}$. This structural correspondence implies the equality of the entire sets: $\partial_{\boldsymbol{\eta}}^\circ \tilde{\Phi}(T\boldsymbol{\theta}; \boldsymbol{x}) = T(\partial_{\boldsymbol{\theta}}^\circ \Phi(\boldsymbol{\theta}; \boldsymbol{x}))$. $\qquad\square$

## C Proofs from section 5

### C.1 Proof of theorem 5.2

We aim to show that if $\boldsymbol{\theta}(t)$ solves $\frac{d\boldsymbol{\theta}}{dt} \in -\partial^\circ \mathcal{L}(\boldsymbol{\theta})$, then $\boldsymbol{\eta}(t) = T\boldsymbol{\theta}(t)$ solves $\frac{d\boldsymbol{\eta}}{dt} \in -\partial^\circ \tilde{\mathcal{L}}(\boldsymbol{\eta}(t))$. We have $\frac{d\boldsymbol{\eta}}{dt} = T\frac{d\boldsymbol{\theta}}{dt}$. We need to show $T(-\partial^\circ \mathcal{L}(\boldsymbol{\theta}(t))) = -\partial^\circ \tilde{\mathcal{L}}(T\boldsymbol{\theta}(t))$, which is equivalent to $T(\partial^\circ \mathcal{L}(\boldsymbol{\theta})) = \partial^\circ \tilde{\mathcal{L}}(T\boldsymbol{\theta})$.

Using the chain rule (valid under Assumption 5.1(A3) as $\ell$ is $\mathcal{C}^1$-smooth) and properties of $T$ (network equivalence $\tilde{\Phi}(T\boldsymbol{\theta}; \boldsymbol{x}) = \Phi(\boldsymbol{\theta}; \boldsymbol{x})$ and subgradient equality $\partial_{\boldsymbol{\eta}}^\circ \tilde{\Phi}(T\boldsymbol{\theta}; \boldsymbol{x}) = T(\partial_{\boldsymbol{\theta}}^\circ \Phi(\boldsymbol{\theta}; \boldsymbol{x}))$ from Theorem 4.8 (which relies on Assumptions 4.1(A1, A2)) or Theorem A.4 as appropriate):

$$
\begin{aligned}
\partial^\circ \tilde{\mathcal{L}}(T\boldsymbol{\theta}) &= \sum_{k=1}^n \ell'(y_k \tilde{\Phi}(T\boldsymbol{\theta}; \boldsymbol{x}_k)) y_k \partial_{\boldsymbol{\eta}}^\circ \tilde{\Phi}(T\boldsymbol{\theta}; \boldsymbol{x}_k) \\
&= \sum_{k=1}^n \ell'(y_k \Phi(\boldsymbol{\theta}; \boldsymbol{x}_k)) y_k [T(\partial_{\boldsymbol{\theta}}^\circ \Phi(\boldsymbol{\theta}; \boldsymbol{x}_k))] \quad \text{(Using Thm. 4.8 or A.4 properties)} \\
&= T\left[ \sum_{k=1}^n \ell'(y_k \Phi(\boldsymbol{\theta}; \boldsymbol{x}_k)) y_k \partial_{\boldsymbol{\theta}}^\circ \Phi(\boldsymbol{\theta}; \boldsymbol{x}_k) \right] \quad \text{(by linearity of } T \text{ and sum rule)} \\
&= T(\partial^\circ \mathcal{L}(\boldsymbol{\theta})).
\end{aligned}
$$

Thus $T(-\partial^\circ \mathcal{L}(\boldsymbol{\theta})) = -\partial^\circ \tilde{\mathcal{L}}(T\boldsymbol{\theta})$. If $\frac{d\boldsymbol{\theta}}{dt} \in -\partial^\circ \mathcal{L}(\boldsymbol{\theta}(t))$, then $\frac{d\boldsymbol{\eta}}{dt} = T\frac{d\boldsymbol{\theta}}{dt} \in T(-\partial^\circ \mathcal{L}(\boldsymbol{\theta}(t))) = -\partial^\circ \tilde{\mathcal{L}}(T\boldsymbol{\theta}(t)) = -\partial^\circ \tilde{\mathcal{L}}(\boldsymbol{\eta}(t))$. With matching initial conditions $\boldsymbol{\eta}(0) = T\boldsymbol{\theta}(0)$, and assuming unique solutions exist for the gradient flow (as per Assumption 5.1(A4)), the trajectories coincide: $\boldsymbol{\eta}(t) = T\boldsymbol{\theta}(t)$ for all $t \geq 0$. $\qquad\square$

### C.2 Proof of theorem 5.4

We need to show two inclusions: $T(L(\boldsymbol{\theta}(0))) \subseteq L(\boldsymbol{\eta}(0))$ and $L(\boldsymbol{\eta}(0)) \subseteq T(L(\boldsymbol{\theta}(0)))$. This proof requires Assumptions 4.1(A1, A2) and 5.1(A3, A4).

1. **Show** $T(L(\boldsymbol{\theta}(0))) \subseteq L(\boldsymbol{\eta}(0))$**:** Let $\boldsymbol{x} \in L(\boldsymbol{\theta}(0))$. By Definition 5.3, there exists a sequence $t_k \to \infty$ such that the normalized trajectory $\bar{\boldsymbol{\theta}}(t_k) = \boldsymbol{\theta}(t_k)/\|\boldsymbol{\theta}(t_k)\|_2$ converges to $\boldsymbol{x}$. Note that $\|\boldsymbol{\theta}(t_k)\|_2 \to \infty$ by Assumption 5.1(A4), so normalization is well-defined for large $k$. From Theorem 5.2, we have $\boldsymbol{\eta}(t) = T\boldsymbol{\theta}(t)$. Since $T$ is an isometry (proven in Theorems A.4, 4.8), $\|\boldsymbol{\eta}(t)\|_2 = \|T\boldsymbol{\theta}(t)\|_2 = \|\boldsymbol{\theta}(t)\|_2$. Therefore, the normalized trajectories are related by:

$$
\bar{\boldsymbol{\eta}}(t) = \frac{\boldsymbol{\eta}(t)}{\|\boldsymbol{\eta}(t)\|_2} = \frac{T\boldsymbol{\theta}(t)}{\|\boldsymbol{\theta}(t)\|_2} = T\left( \frac{\boldsymbol{\theta}(t)}{\|\boldsymbol{\theta}(t)\|_2} \right) = T(\bar{\boldsymbol{\theta}}(t)).
$$

Now consider the sequence $\bar{\boldsymbol{\eta}}(t_k) = T(\bar{\boldsymbol{\theta}}(t_k))$. Since $T$ is a linear transformation in finite-dimensional spaces, it is continuous. As $k \to \infty$, $\bar{\boldsymbol{\theta}}(t_k) \to \boldsymbol{x}$ implies $T(\bar{\boldsymbol{\theta}}(t_k)) \to T(\boldsymbol{x})$. So, we have found a sequence $t_k \to \infty$ such that $\bar{\boldsymbol{\eta}}(t_k) \to T(\boldsymbol{x})$. By the definition of the $\omega$-limit set, this means $T(\boldsymbol{x}) \in L(\boldsymbol{\eta}(0))$. Since this holds for any arbitrary $\boldsymbol{x} \in L(\boldsymbol{\theta}(0))$, we conclude $T(L(\boldsymbol{\theta}(0))) \subseteq L(\boldsymbol{\eta}(0))$.

2. **Show** $L(\boldsymbol{\eta}(0)) \subseteq T(L(\boldsymbol{\theta}(0)))$**:** Let $\boldsymbol{y} \in L(\boldsymbol{\eta}(0))$. By Definition 5.3, there exists a sequence $t_k' \to \infty$ such that $\bar{\boldsymbol{\eta}}(t_k') \to \boldsymbol{y}$. We know $\bar{\boldsymbol{\eta}}(t_k') = T(\bar{\boldsymbol{\theta}}(t_k'))$. The sequence $\{\bar{\boldsymbol{\theta}}(t_k')\}_{k=1}^\infty$ lies on

the unit sphere $\mathbb{S}^{m-1}$ in $\mathbb{R}^m$, which is a compact set. By the Bolzano-Weierstrass theorem, there must exist a convergent subsequence. Let $\{t'_{k_j}\}_{j=1}^{\infty}$ be the indices of such a subsequence, so that $\bar{\boldsymbol{\theta}}(t'_{k_j}) \to \boldsymbol{x}'$ for some $\boldsymbol{x}' \in \mathbb{S}^{m-1}$ as $j \to \infty$. Since $t'_{k_j} \to \infty$ as $j \to \infty$, the limit point $\boldsymbol{x}'$ must belong to the $\omega$-limit set of the original trajectory $\bar{\boldsymbol{\theta}}(t)$, i.e., $\boldsymbol{x}' \in L(\boldsymbol{\theta}(0))$. Now consider the corresponding subsequence for $\bar{\boldsymbol{\eta}}$: $\{\bar{\boldsymbol{\eta}}(t'_{k_j})\}_{j=1}^{\infty}$. Since $T$ is continuous, as $j \to \infty$, $\bar{\boldsymbol{\theta}}(t'_{k_j}) \to \boldsymbol{x}'$ implies $T(\bar{\boldsymbol{\theta}}(t'_{k_j})) \to T(\boldsymbol{x}')$. So, $\bar{\boldsymbol{\eta}}(t'_{k_j}) \to T(\boldsymbol{x}')$. However, the original sequence $\bar{\boldsymbol{\eta}}(t'_k)$ converges to $\boldsymbol{y}$. Any subsequence of a convergent sequence must converge to the same limit. Therefore, we must have $\boldsymbol{y} = T(\boldsymbol{x}')$. Since $\boldsymbol{y}$ was an arbitrary element of $L(\boldsymbol{\eta}(0))$, and we found an element $\boldsymbol{x}' \in L(\boldsymbol{\theta}(0))$ such that $\boldsymbol{y} = T(\boldsymbol{x}')$, this demonstrates that every element in $L(\boldsymbol{\eta}(0))$ is the image under $T$ of some element in $L(\boldsymbol{\theta}(0))$. Hence, $L(\boldsymbol{\eta}(0)) \subseteq T(L(\boldsymbol{\theta}(0)))$.

Combining both inclusions yields $T(L(\boldsymbol{\theta}(0))) = L(\boldsymbol{\eta}(0))$. $\qquad\square$

### C.3 Proof of corollary 5.5

If the normalized trajectory $\bar{\boldsymbol{\theta}}(t)$ converges to a unique limit $\overline{\boldsymbol{\theta}}^*$, then by Definition 5.3, its $\omega$-limit set is $L(\boldsymbol{\theta}(0)) = \{\overline{\boldsymbol{\theta}}^*\}$. Under the specified assumptions in Corollary 5.5 (which include Assumptions 4.1(A1, A2), Assumption 5.1(A3), and trajectory properties (A4)), Theorem 5.4 states that $T(L(\boldsymbol{\theta}(0))) = L(\boldsymbol{\eta}(0))$. Substituting $L(\boldsymbol{\theta}(0))$, we have $L(\boldsymbol{\eta}(0)) = T(\{\overline{\boldsymbol{\theta}}^*\}) = \{T\overline{\boldsymbol{\theta}}^*\}$. Since the $\omega$-limit set $L(\boldsymbol{\eta}(0))$ consists of the single point $T\overline{\boldsymbol{\theta}}^*$, this implies that the normalized trajectory $\bar{\boldsymbol{\eta}}(t)$ converges to $T\overline{\boldsymbol{\theta}}^*$ as $t \to \infty$. $\qquad\square$

## D  Experimental Details

To validate our theoretical results on trajectory preservation (Theorem 5.2), we conduct a series of experiments corresponding to the results summarized in Table 1 and Table 2. We use PyTorch for all implementations. The core methodology involves training a pair of networks—a narrow network $\Phi$ and its wider counterpart $\tilde{\Phi}$ constructed via our transformation $T$—and measuring the trajectory error $\|\boldsymbol{\eta}(t) - T\boldsymbol{\theta}(t)\|_2$ at each step.

### D.1  Experiments on 2D Toy Datasets

**Dataset Generation.** We use a two-dimensional toy dataset consisting of 100 samples from two classes ($\pm 1$). For the separable case (Exp. 1, 3, 4, 5), the data points are generated from two Gaussian distributions centered at $(2, -2)$ and $(-2, 2)$, ensuring they are separable by a line passing through the origin. For the non-separable case (Exp. 2), the distributions are centered closer to the origin at $(1, -1)$ and $(-1, 1)$ to create overlap.

**Model Architectures.** We test both fully-connected (MLP) and convolutional (CNN) networks. All models are homogeneous, using Leaky ReLU with a negative slope of $\alpha = 0.01$ as the activation function. Weights are initialized using Kaiming Normal/Uniform initialization as detailed in the code.

- **MLP** (Exp. 1-4): The narrow network is a 2-layer MLP with a single hidden neuron (Input(2) $\to$ Hidden(1) $\to$ Output(1)). The wide network splits this to two hidden neurons (Input(2) $\to$ Hidden(2) $\to$ Output(1)).
- **CNN** (Exp. 5): The narrow network takes a $1 \times 5 \times 5$ input and consists of a convolutional layer with 2 output channels (3x3 kernel, stride 1, no padding), followed by an average pooling layer (2x2 kernel, stride 2), a flatten operation, and a final linear layer to produce a single output. The wide network splits the convolutional layer to 4 channels, adjusting the subsequent linear layer input dimension accordingly.

**Training Setup.** Networks are trained for $100,000$ steps using an exponential loss function. The learning rate is 0.1 for MLP experiments and 0.001 for the CNN experiment. For Gradient Descent (GD, Exp. 1, 2, 5), the full batch is used in each step. For Stochastic Gradient Descent (SGD, Exp. 3, 4), we use a batch size of 16. Crucially, for the identical batch experiment (Exp. 3), both networks

are fed the exact same sequence of mini-batches, whereas for the different batch experiment (Exp. 4), they use independent data loaders. The initial parameters of the wide network are always set to $\boldsymbol{\eta}(0) = T\boldsymbol{\theta}(0)$ using splitting coefficients $c_i = 1/\sqrt{m_{split}}$.

**Results.** The numerical results for maximum trajectory error are summarized in Table 1. Figures 3, 4, and 5 provide visual confirmation and further details. The error remains near machine precision ($\sim 10^{-13}$) in all cases where the theory predicts preservation (Exp. 1, 2, 3, 5), regardless of data separability or the use of SGD (with identical batches). In contrast, when the SGD batch order differs (Exp. 4), the error grows substantially, confirming the necessity of identical training paths. The slightly higher error floor for the CNN (Exp. 5, $\sim 10^{-10}$) is consistent with the expected accumulation of floating-point errors due to the higher computational complexity of convolutions.

Table 1: Complete results for maximum trajectory error on 2D toy datasets.

| Exp. | Model | Optimizer | Data Condition | Max Trajectory Error |
|---|---|---|---|---|
| 1 | MLP | GD | Separable | $4.46 \times 10^{-13}$ |
| 2 | MLP | GD | Non-Separable | $5.57 \times 10^{-13}$ |
| 3 | MLP | SGD (Identical batches) | Separable | $4.35 \times 10^{-13}$ |
| 4 | MLP | SGD (Different batches) | Separable | $1.53 \times 10^{-1}$ |
| 5 | CNN | GD | Separable | $2.36 \times 10^{-10}$ |

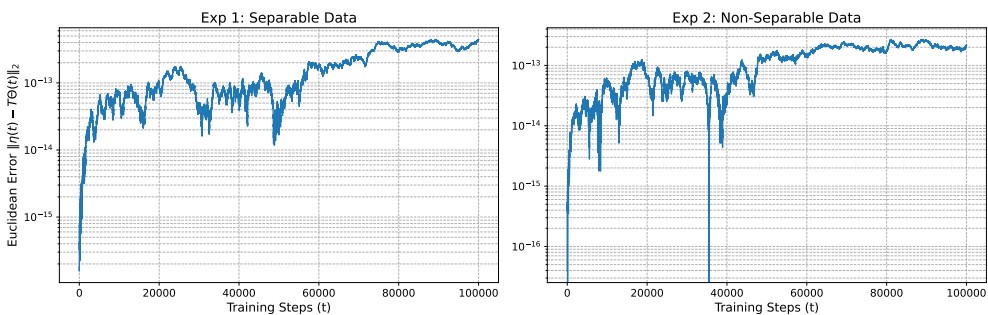

Figure 3: Trajectory error comparison for MLP experiments using Gradient Descent (GD). (Left) Exp. 1 on separable data shows error near machine precision. (Right) Exp. 2 on non-separable data also shows error remaining near machine precision, demonstrating robustness to data conditions.

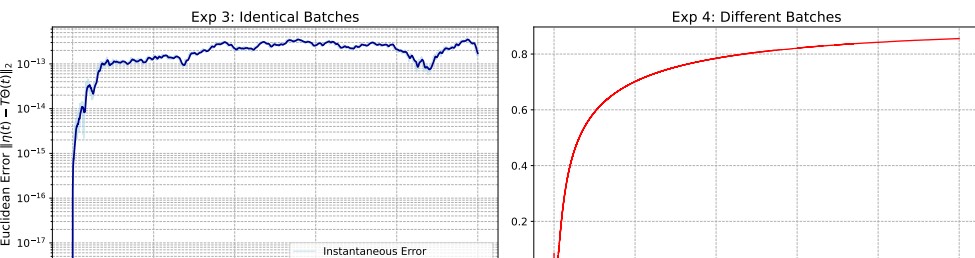

Figure 4: Trajectory error comparison for MLP experiments using Stochastic Gradient Descent (SGD). (Left) Exp. 3, using identical mini-batches for both networks, maintains error near machine precision (moving average shown). (Right) Exp. 4, using different mini-batches, shows substantial error growth, highlighting the necessity of identical data sequences.

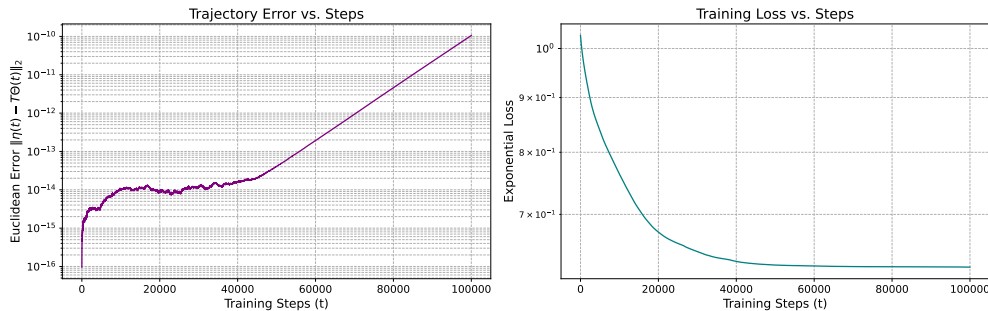

Figure 5: Results for the CNN experiment using GD on separable data (Exp. 5). (Left) The trajectory error remains small ($< 10^{-9}$) but exhibits a slight upward trend, consistent with accumulated precision errors. (Right) The training loss converges successfully.

## D.2 Experiment on the MNIST Dataset

**Dataset and Models.** To test our principle on a more realistic task, we use the MNIST dataset, focusing on the binary classification of digits '3' versus '5'. We use larger homogeneous models: an MLP splitting from 128 to 256 hidden units, and a CNN splitting from 10 to 20 channels (detailed architectures follow standard practices, similar to the toy CNN but scaled appropriately for MNIST image size).

**Training Setup.** Both models are trained for 1000 epochs using SGD with a learning rate of $0.001$ and a batch size of $64$. Data is normalized. Identical mini-batches are used for the narrow and wide networks at each step.

**Results.** The trajectory preservation principle continues to hold with remarkable precision on this practical task. The maximum trajectory errors, summarized in Table 2, remain close to machine precision for both architectures. Figures 6 and 7 provide a visual representation of the dynamics. The left panels show the trajectory error remaining consistently low throughout the 1000 epochs. The right panels display the corresponding training loss (evaluated on the full training set), confirming that both models learned effectively, achieving high final classification accuracies (MLP: 99.99%, CNN: 99.82%). These results demonstrate the principle's validity on a standard dataset with larger models.

Table 2: Maximum trajectory error on the MNIST dataset.

| Model Configuration | Max Trajectory Error |
|---|---|
| MLP ($128 \rightarrow 256$ units) | $2.19 \times 10^{-13}$ |
| CNN ($10 \rightarrow 20$ channels) | $1.28 \times 10^{-13}$ |

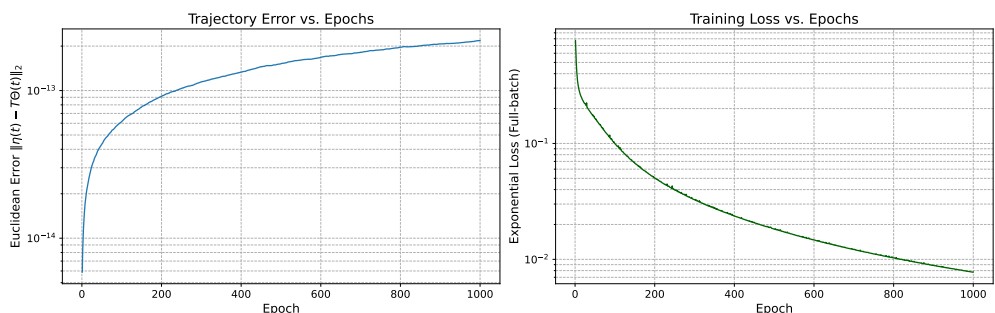

Figure 6: MNIST results for MLP (128 → 256 units). **(Left)** Trajectory error per epoch remains near machine precision. **(Right)** Training loss (full-batch evaluation) converges successfully.

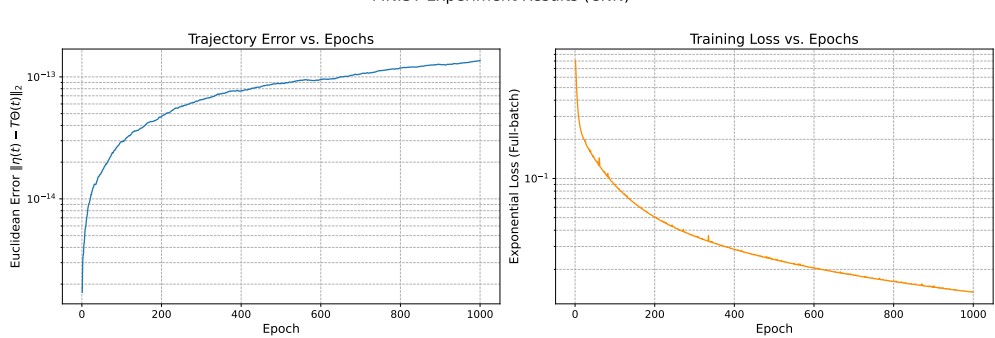

Figure 7: MNIST results for CNN (10 → 20 channels). **(Left)** Trajectory error per epoch remains near machine precision. **(Right)** Training loss (full-batch evaluation) converges successfully.

