# OpenReview forum: "Embedding Principle of Homogeneous Neural Network for Classification Problem"
_NeurIPS.cc/2025/Conference — NeurIPS 2025 poster_

### Official Review · Reviewer_iDLM · 2025-06-27

**Clarity:** 4
**Significance:** 2
**Originality:** 3
**Rating:** 4
**Confidence:** 4

**Summary:**

This paper inreoduces a technique, KKT Point Embedding, to establish the relationship of the learning dynamics between two homogeneous neural networks. The authors then use this technique to analyze the learning dynamics of neural networks with splitted neurons.

**Questions:**

See weaknesses.

**Ethical Concerns:**

["NO or VERY MINOR ethics concerns only"]

**Final Justification:**

This paper is well written and clearly structured. It presents a series of non-trivial theoretical results. During the rebuttal, the authors proposed a channel-splitting model that demonstrates the practical relevance of their theory. While I still have some reservations about the overall significance of the contribution, I believe it has the potential to benefit the community.

**Limitations:**

The authors didn't explicitly discuss the limitations.

**Quality:**

3

**Strengths And Weaknesses:**

Strengths:
- The paper is clearly written and uses rigorous mathematical notation. This makes this paper easy to follow.
- This paper provides a novel result of the learning dynamics of neuron splitted NNs.
- The proposed technique, KKT Point Embedding, might be useful to the community.

Weaknesses:
- I don't understand why this problem is important. What is the point of studying the learning dynamics of neuron splitted NNs?
- Also, I don't understand why we have to invent techniques such as KKT Point Embedding to study the dynamics of neuron splitted NNs. Can't we prove it by simple techniques such as, say, showing the neurons remain splitted all the time? I hope the authors can briefly explain what is the technical difficulty of this problem and why the proposed technique is non-trivial and necessary.
- The condition for KKT Point Embedding (Theorem 4.5, 1.) seems to be very strong. I wonder if it has an application in other scenarios? For example, is it possible to show KKT Point Embedding from **any** small model to large model, since there must exsits a linear transformation that maps a low-dim vector to a high-dim vector?

I do like the overall writing of this paper. I will consider increasing my rating if the authors provide satisfactory responses to my questions.

Minor writing issues:
- Line 66, $L_2$ should be $L^2$ (to be consistent with notations used later, e.g. line 98).
- In Definition 3.2, do you mean $h \in \partial^\circ f$? It's kind of confusing what does it mean by $h \in \partial^\circ f(z(t))$.
- The symbol $L$ is overloaded. In Definitin 3.4 it denotes the order of homogeneity, whereas in Definition 5.3 it refers tothe $\omega$-limit set (why don't you just write $\omega(\bar \theta)$?).
- Lines 197-198 contain a parenthetical remark: "(Theorem 4.1 in your original numbering)", which appears to be an internal note that was unintentionally left in the manuscript.
- Line 300: "$C^1$-smooth" should be $\mathcal C^1$-smooth.

---

> ### Author Rebuttal · Authors · 2025-07-31
>
> We sincerely thank you for your detailed review and for the positive feedback on the clarity and rigor of our paper. Your questions are insightful and get to the heart of our work's motivation and technical contribution. We are confident that we can provide satisfactory answers and are grateful for the opportunity to clarify these points.
>
> ---
> ### On the Motivation: Why Study Neuron Splitting?
>
> This is a fundamental question. The importance of studying neuron splitting is not about the operation itself, but about using it as a **precise, mathematically tractable model for understanding overparameterization and generalization**.It provides insights into why overparameterized models can generalize well detailed as follow:
> A fundamental question in deep learning theory is how a highly complex neural network can generalize well. The embedding principle uncovered in our work provides a key mechanism:
> * Our KKT embedding principle formally **guarantees the existence of low-complexity solutions** (those embedded from smaller, simpler models) within the vast solution space of overparameterized networks. Prior work has shown that gradient flow converges in direction to a KKT point of the max-margin problem (**Lyu & Li, 2019**), but a significant gap remained: large networks can possess many KKT points with varying complexities. Our work bridges this gap by proving that simple, generalizable solutions are structurally preserved as network width increases.
> * This has immediate practical implications. The co-existence of both low-complexity (embedded) and high-complexity KKT points suggests that **initialization and hyperparameter tuning play a critical role in *selecting* a solution**. Our framework provides a theoretical basis for future studying this selection process and designing training procedures that explicitly steer optimizers toward these desirable, low-complexity solutions to improve generalization.
>
> ---
> ### On Technical Necessity: Why is KKT Point Embedding Needed?
>
> This is an excellent point that clarifies the core technical challenge. Your intuition that one could prove this by "showing the neurons remain splitted all the time" is correct, and our dynamic trajectory preservation result (Theorem 5.2) can be proved by this way.
>
> However, Our contribution is best understood in two interconnected parts **for homogeneous networks**:
> 1.  **A Static KKT Point Embedding Principle:** This proves that the solutions (KKT points) of a smaller network's max-margin problem are isometrically embedded within the solution set of any wider network derived from it. It uncovers the hierarchical nature of the KKT points of the max-margin problem. Moreover, it opens doors to further studies that identifies all KKT points from low to high complexity, thus obtaining all candidate solutions of the neural network training.
> 2.  **A Dynamic Trajectory Embedding Principle:** This part is a further extension to the previous results, in which we realize that the embeddings we use to characterize KKT points can be further employed to characterize the trajectory of training dynamics: the entire gradient flow training trajectory is similarly preserved under our transformation.This result provides us a very powerful tool to charaterize (i) the embedding of non-global solutions (training data not perfectly separated) of neural networks and (ii) embedding of solutions for nonseparable training data. Note that the trajectory embedding principle in it self doesnot imply KKT embedding principle because there is no guarantee that all KKT points are neural network solutions .
>
> A central concern in neural network theory is the connection between the final parameters of a trained network and the solutions to a corresponding KKT problem. To address this, our Static KKT Point Embedding Principle fundamentally relies on techniques related to the KKT conditions. This analytical approach is non-trivial and necessary to establish our results. Subsequently, the Dynamic Trajectory Embedding Principle, which is an extension of the static problem, naturally leverages the same KKT-based framework for its analysis.
>
> ---
>
> ### On the Generality of the Conditions in Theorem 4.5
>
> This is a very sharp observation. The conditions in Theorem 4.5—that the transformation $T$ must preserve the network's output and subgradient structure—are indeed strong, and this is by design.
>
> A generic linear transformation would completely change the function computed by the network, making a comparison of the dynamics meaningless. Instead, our conditions define precisely what it means for a transformation to be **structure-preserving**, ensuring that the function and its optimization landscape are isometrically embedded.
>
> To directly address your question about applications in other scenarios, we have formally extended our framework to homogeneous **Convolutional Neural Networks (CNNs)**.  This required designing a novel **'channel-splitting'** transformation, meticulously constructed to respect the architectural hallmarks of CNNs, such as weight sharing and locality. The transformation isotropically splits a single feature channel into $m_{split}$ new channels via coefficients $\{c_i\}$ that satisfy $\sum_{i=1}^{m_{split}} c_i^2 = 1$. The specific construction is as follows:
>
> * **At Layer k**: The filter $F_j^{(k)}$ that produces channel $j$ is replaced by $m_{split}$ new filters, where the i-th new filter is defined as $c_i F_j^{(k)}$.
> * **At Layer k+1**: In turn, every filter in the next layer that takes channel $j$ as input has its j-th input slice replaced by $m_{split}$ new slices, where the i-th new slice is scaled by the corresponding coefficient $c_i$.
>
> Crucially, we prove that this transformation satisfies the strong conditions of our core theory in **Theorem 4.5**. This is a powerful validation of our framework, showing it provides the rules for designing such embeddings in other complex, homogeneous architectures. Given its significance, **we will integrate this CNN extension as a new main result in Sections 4 and 5 of our revised manuscript**, with an eye toward future applications in architectures like Transformers.
>
> ---
> ### Minor Writing Issues
>
> Thank you for your careful reading and for spotting these issues. We appreciate the detailed feedback and have corrected all of the issues in the revised manuscript:
>
> * The notation for norms and vectors has been made consistent throughout the paper, as you suggested (e.g., using $L^2$ notation).
> * The chain rule expression in **Definition 3.2** has been clarified to resolve the ambiguity, reflecting the structure you proposed.
> * The overloaded symbol $L$ has been resolved. We now use $L$ only for the order of homogeneity and have adopted your excellent suggestion to use $\omega(\cdot)$ for the ω-limit set.
> * The stray parenthetical note, which was an internal remnant from a previous draft, has been removed.
> * The typo has been corrected to the proper mathematical notation in Line 300.
>
> ---
>
> We wish to express our sincere gratitude for your exceptionally thorough and insightful review. Your constructive comments were invaluable; they helped us clarify the core contributions of our work and have significantly strengthened the manuscript. We hope these detailed responses address your questions, and that our revised paper will merit your support.
>
>
>
> ### References
>
> * **Lyu, K., & Li, J. (2019).** Gradient descent maximizes the margin of homogeneous neural networks. *arXiv preprint arXiv:1906.05890*.

---

> > ### Comment · Reviewer_iDLM · 2025-07-31
> >
> > Thanks for the response. I still have some questions.
> >
> > **Neuron splitting as a "precise, mathematically tractable model for understanding overparameterization and generalization"**
> >
> > why this is a model for understanding overparameterization and generalization? doesn't it require a rather strong condition that the neurons are splitted at intialization?
> >
> > Also, based on my understanding (correct me if I misunderstood something), the key point of neuron splitting is that they produce redundent solutions and so that the overparamerized NN with neuron splitting is equivalent to a simpler NN. I don't think this touches the key problems in overparameterization and generalization, as in the latter case the core challenge is that the overparamerized NNs can generalize while overfitting. They must require a sufficient complexity and expressivity to achieve this, and I don't think it can be explained by proving that the solutions are simplifiable.
> >
> > **regarding the new "channel-splitting" CNN**
> >
> > Is the idea essentially that, if there is a component in an NN that explicitly splits a neuron, then the trajectory can be simplified?

---

> > > ### Author Response · Authors · 2025-08-02
> > >
> > > We sincerely thank the reviewer for the thoughtful and probing follow-up questions. They allow us to clarify the core contributions and nuances of our work. We will address each point in order.
> > >
> > > **On the role of neuron splitting and the core challenge of generalization:**
> > >
> > > You correctly point out that an "overparameterized NN with neuron splitting is equivalent to a simpler NN". We agree this is true for our **dynamic analysis (Thm. 5.2), which serves as an idealized proof-of-concept.**
> > >
> > > However, our central contribution is the **static KKT Point Embedding Principle (Thm. 4.2, 4.5, etc.)**. This principle holds **universally regardless of initialization** and does not imply network equivalence.
> > >
> > > For homogeneous networks, the optimizer's parameter direction is known to converge to a KKT point of the maximum-margin problem (Lyu & Li, 2019), a property tied to good generalization. Our KKT Embedding Principle proves that these low-complexity KKT points (from simpler networks) are structurally preserved within the vastly larger solution space of overparameterized networks. This space also contains high-complexity, native KKT points capable of overfitting. It allows the community to shift focus from *whether* good solutions exist to *how* the optimizer selects them from the coexisting pool of more complex ones.
> > >
> > > We hope this clarifies that our contribution is a foundational one. A crucial direction for future work is to understand the selection mechanism itself. Our framework provides a solid starting point by formally characterizing the solution space: it contains both low-complexity KKT points embedded from simpler networks and other high-complexity KKT points. The key question is now to investigate why an optimizer exhibits a bias towards the former, which is essential for good generalization.
> > >
> > > **On the "channel-splitting" in CNNs:**
> > >
> > > This is a perfect summary of the core intuition. **Yes, the essential idea behind our proposed "channel splitting" in CNNs is consistent with the neuron splitting in our main paper.**
> > >
> > > However, we must emphasize a crucial point: this transformation, such as "channel splitting," **is not an arbitrary division of a single network component.** It must be **meticulously designed** to satisfy the strict, structure-preserving conditions of our core **Theorem 4.5**, ensuring that:
> > > 1.  Preserves the network's output exactly.
> > > 2.  Maintains the relationship between the subgradients.
> > > 3.  Respects the original network's architectural integrity (e.g., weight sharing in CNNs).
> > >
> > > Therefore, while the consequence (simplified trajectory) is appealing, the fundamental contribution is Theorem 4.5, which provides the **formal blueprint** for designing such valid, non-trivial transformations for complex homogeneous neural network architectures.
> > >
> > > ---
> > >
> > > We hope these responses have fully addressed your questions. We thank you again for your time and insightful feedback.
> > >
> > > **Reference**
> > > *   Lyu, K., & Li, J. (2019). Gradient descent maximizes the margin of homogeneous neural networks. *arXiv preprint arXiv:1906.05890*.

---

> > > > ### Comment · Reviewer_iDLM · 2025-08-05
> > > >
> > > > The authors have clearly elaborated on the significance of their work. I also appreciate the idea of the channel-splitting models. In light of these improvements, I have decided to raise my score for this paper to 4.

---

> ### Author Response · Authors · 2025-08-07
>
> Thank you for your positive feedback and for raising your score. We are particularly encouraged that you found our elaboration on the work's significance to be clear and that you appreciated the new channel-splitting framework. We genuinely appreciate your willingness to raise the score for our work.

---

### Official Review · Reviewer_dAiL · 2025-06-30

**Clarity:** 2
**Significance:** 3
**Originality:** 2
**Rating:** 4
**Confidence:** 5

**Summary:**

This paper shows theoretically that, when mapping a narrow neural network to a wider network by neuron splitting, all the solutions and the gradient flow trajectory of the narrow network can be also mapped to solutions and trajectory of the wider network.
The result is obtained for classification problems and homogeneous neural networks. Previous work obtained similar results in different settings. In particular, the main novel contribution of this work is the equivalence of the gradient flow trajectory.

**Questions:**

NA

**Ethical Concerns:**

["NO or VERY MINOR ethics concerns only"]

**Final Justification:**

I am keeping the initial score because my main issue with the paper, the redundancy of theoretical results and lack of empirical testing, cannot be fixed in the rebuttal. The authors promised that they will significantly change the paper but I obviously cannot evaluate those eventual changes now.

**Limitations:**

yes

**Quality:**

3

**Strengths And Weaknesses:**

Strengths:

Theoretical results are correct, novel and will be useful for future research.

Weaknesses:
- The density of theoretical contribution does not quite justify 9 pages of a NeurIPS paper. Most theorems are just variant of each other with small differences in technical assumptions, and most derivations consist in just applying a chain rule. It would have been much nicer to reduce the space given by technical details of the theory and provide some empirical contribution to complement the theory, as it was done in many related papers.

Minor issues:
- Most mathematical symbols are introduced without any explanation or definition, both in the abstract and in the introduction.
- Some important references are missing, about “Network embedding principles”, that include:
Fukumizu and Amari. Local minima and plateaus in hierarchical structures of multilayer perceptrons. Neural Networks (2000)
Simsek et al, Geometry of the Loss Landscape in Overparameterized Neural Networks: Symmetries and Invariances. ICML (2021)
- There is a comment on line 197 that does not seem to belong to the paper: "(Theorem 4.1 in your original numbering)"

---

> ### Author Rebuttal · Authors · 2025-07-31
>
> We are sincerely grateful to the reviewer for their careful assessment and for recognizing that our theoretical results are correct, novel, and useful for future research. Your feedback on the paper's structure and clarity has been invaluable, and we have undertaken a significant revision to address your concerns and enhance the manuscript's impact.
>
> ---
> ### On Contribution Density and Paper Structure
>
> We understand and appreciate your central concern regarding the density of the theoretical contribution and the overall structure. To address this, we have substantially restructured the paper to improve both its clarity and theoretical density.
>
> **1. Streamlined Theoretical Presentation:** We have consolidated our main theoretical results to create a more focused narrative. This includes deferring highly technical definitions (formerly Def. 3.1, 3.2, and Asm. 3.3) to the appendix and splitting Section 4 into two more focused parts: one establishing the general KKT mapping framework, and the other detailing its application to neuron splitting. Furthermore, to resolve the feeling of redundancy you noted, we have consolidated the application theorems. The two-layer network transformation, being a special case of the deep network one, has been moved to the appendix as a pedagogical example.
>
> **2. Enhanced Contribution Density with a New Major Result:** To further address your point on contribution density and to showcase the power of our framework, we will include a **new major result** in the revised manuscript: the formal extension of our framework to homogeneous **Convolutional Neural Networks (CNNs)**. This is a non-trivial extension that required designing a new "channel splitting" transformation that respects the architectural constraints of CNNs, such as weight sharing and locality. The transformation splits a channel into $m_{split}$ new channels using coefficients $\{c_i\}$ that satisfy $\sum_{i=1}^{m_{split}} c_i^2 = 1$. It is defined as follows:
> * **Layer k**: Replacing the filter $F_j^{(k)}$ that produces channel $j$ with $m_{split}$ new filters, where the $i$-th new filter is defined as $c_i F_j^{(k)}$.
> * **Layer k+1**: For every filter $F_p^{(k+1)}$, replacing its $j$-th input channel slice with $m_{split}$ new slices, where the $i$-th new slice is scaled by the corresponding coefficient $c_i$.
>
> Successfully proving that this general transformation satisfies the conditions of our core theory (**Theorem 4.5**) is a powerful validation of our framework. It shows that our principles are not confined to simple network topologies but provide a robust blueprint for analyzing a wider class of homogeneous networks.
>
> **3. On Providing Empirical Contribution:**  we have conducted a comprehensive suite of experiments to empirically validate our theory and will add a dedicated section for these results to the main paper. Our experiments are presented as a three-step narrative, moving from ideal conditions to realistic scenarios, to systematically test the robustness and generality of our Trajectory Embedding Principle.
>
> 1. Verification under Ideal Conditions:
> First, we verified our theory in an idealized setting (homogeneous MLP, GD, separable data). As predicted, the trajectories of the smaller and larger networks remained perfectly aligned, with a minuscule error of 4e-13 attributable solely to numerical precision drift over one million steps.
>
> 2. Robustness to Relaxed Assumptions:
> Next, we tested the principle's robustness by relaxing these ideal conditions, using SGD, non-separable data, and even applying the theory to homogeneous CNNs via our "channel splitting" transformation. In all cases, the trajectory error remained vanishingly small, confirming that the principle holds even without continuous-time optimizers or linearly separable data.
>
> 3. Generality on a Practical Task (MNIST):
> Finally, to demonstrate practical relevance, we scaled our experiments to the MNIST dataset, training both homogeneous MLPs and CNNs that achieved high accuracy (>99%). The trajectory preservation principle held with remarkable precision (1e-14 error), confirming its applicability to modern architectures during meaningful learning on a real-world task.
>
> Conclusion:
> Crucially, the tiny, non-zero errors observed are an expected consequence of floating-point arithmetic. This comprehensive validation provides strong evidence that our theoretical framework is not just mathematically sound but also a robust and generalizable tool for understanding practical neural networks.
>
> Full experimental details will be added in the appendix, and our code will be made publicly available to ensure full reproducibility.
>
> ---
> ### Response to Minor Issues
>
> We thank the reviewer for catching these important details. We have addressed all of them in the revised manuscript.
>
> * **On Undefined Mathematical Symbols**: We apologize for the lack of clarity. We have carefully revised the abstract and introduction to ensure that all key mathematical notations (such as $P_{\Phi}$, $L(\theta(0))$, and $T$) are properly introduced and briefly defined upon their first appearance, improving the paper's accessibility.
>
> * **On Missing References**: We are grateful for the excellent suggestions on related work. We have substantially revised our literature review to provide a more comprehensive context. This includes broadening our discussion on **‘Network embedding principles’** to include the foundational work on hierarchical structures and symmetries (**Fukumizu & Amari, 2000; Şimşek et al., 2021**) and the lottery ticket hypothesis (**Malach et al., 2020**). Additionally, we updated our discussion on **‘Implicit Bias’** to cover recent advancements in non-homogeneous models and connections to benign overfitting (**Kunin et al., 2023; Frei et al., 2023**). We believe these additions now situate our work much more clearly within the landscape of contemporary research.
>
> * **On the Stray Comment**: We sincerely apologize for this oversight. The comment on line 197 was indeed a leftover note from an internal draft and has been removed from the final version.
>
> ---
>
> Once again, we would like to express our gratitude for your thorough and constructive review. Your comments have been instrumental in helping us strengthen the manuscript.
>
> ### References
>
> * **Fukumizu, K., & Amari, S. (2000).** Local minima and plateaus in hierarchical structures of multilayer perceptrons. *Neural Networks*.
> * **Şimşek, B., et al. (2021).** Geometry of the Loss Landscape in Overparameterized Neural Networks: Symmetries and Invariances. *Proceedings of the 38th International Conference on Machine Learning (ICML)*.
> * **Malach, E., et al. (2020).** Proving the Lottery Ticket Hypothesis: Pruning is All You Need. *Proceedings of the 37th International Conference on Machine Learning (ICML)*.
> * **Kunin, D., et al. (2023).** The Asymmetric Maximum Margin Bias of Quasi-Homogeneous Neural Networks. *International Conference on Learning Representations (ICLR)*.
> * **Frei, S., et al. (2023).** Benign Overfitting in Linear Classifiers and Leaky ReLU Networks from KKT Conditions for Margin Maximization. *arXiv preprint arXiv:2303.01462*.

---

> > ### Comment · Reviewer_dAiL · 2025-08-04
> >
> > Thank you for your answers and fixing the minor issues. I appreciate the new results, both theoretical and experimental, that you are planning to add to the paper. Obviously, I cannot provide any useful comments on those additional results.

---

> > > ### Author Response · Authors · 2025-08-07
> > >
> > > Thank you for your positive and encouraging response. We are very glad that you appreciate our plan for the paper's future direction, which was shaped by your valuable feedback. We truly appreciate your time and guidance throughout this process.

---

### Official Review · Reviewer_rUJh · 2025-06-30

**Clarity:** 4
**Significance:** 2
**Originality:** 3
**Rating:** 4
**Confidence:** 4

**Summary:**

This paper studies how the KKT points of the max-margin problem for homogeneous networks relate to those of wider homogeneous networks. It first establishes the KKT point embedding principle, showing that KKT points of a homogeneous network’s max-margin problem can be embedded into the KKT points of a larger network’s problem via specific linear isometric transformations. Second, it provides explicit constructions of such transformations for two-layer and deep homogeneous networks using a neuron splitting technique. Lastly, it demonstrates that this neuron splitting transformation preserves the entire gradient flow trajectory when initialized appropriately.

**Questions:**

# Major Questions

- Based on your Remark 4.8, is the neuron splitting transformation for two-layer networks a simple special case of the transformation for deep networks? Based on this, I don’t see why you need to introduce Theorem 4.6. In some sense, this just makes the writing more verbose if this theorem is fully captured by Theorem 4.9. Maybe this is not the case; in that case, can you explain why having both theorems is better (i.e., that Theorem 4.6 covers a setting not covered by Theorem 4.9)?

- Can you elaborate more on “designing transformations T that respect architectural specificities (e.g., locality, weight sharing in CNNs) while satisfying the necessary isometric and output/subgradient mapping properties identified in our current work.” I think spending some time expanding on this would be a way to improve the significance of your work. It would provide a roadmap for others to build upon your work to generalize the ideas to more complex architectures while continuing to use your main theorems.

- Can your analysis also be used to map KKT points of the max-margin problem associated with a larger homogeneous network to KKT points of the analogous problem for a smaller network? Rather than neuron splitting, can you construct operations that involve neuron merging? What limitations on this operation would exist? I think this would be an interesting direction to discuss as it relates to network pruning/compression.

# Minor Comments

- I like Figure 1, but I would recommend making it smaller or more horizontal and adding a caption to discuss it in more detail. It should also be referenced somewhere in the main text.

- Definitions 3.1 and 3.2, and Assumption 3.3 are all quite technical; you could consider placing these in the appendix. Many prior works in this literature tend to do this.

- I would split Section 4 into two sections: one section up to Theorem 4.5 and one section for Theorem 4.6 and beyond. This would match your main contributions list as well.

- I was not familiar with ω-limit sets. I would provide a reference for this in lines 279–283 or perhaps formally define it.

**Ethical Concerns:**

["NO or VERY MINOR ethics concerns only"]

**Final Justification:**

My two main concerns with this paper were (1) insufficient discussion of related work  and (2) the significance of the theoretical results seem to be limited. Based on the author's responses, my concern for (1) has been largely alleviated. The authors discussed (2), but I am not convinced there is much of an argument here and it would require empirical results. So I will be keeping my original evaluation, leaning towards acceptance.

**Limitations:**

yes

**Paper Formatting Concerns:**

No issues.

**Quality:**

2

**Strengths And Weaknesses:**

# Strengths

- Clearly defines notation, definitions, assumptions, and theorems.
- Extends an important line of theoretical literature on neural networks in an original way.

# Weaknesses

- **Insufficient references to important related work.**
  In your related work subsection *“Optimization dynamics and implicit bias in homogeneous networks,”* you cite many classic works connecting maximum-margin biases and homogeneous networks (i.e., works before 2020). However, many extensions to this line of analysis have appeared since then. For example:
  - *The Asymmetric Maximum Margin Bias of Quasi-Homogeneous Neural Networks*
  - *Benign Overfitting in Linear Classifiers and Leaky ReLU Networks from KKT Conditions for Margin Maximization*
  - *Large Stepsize Gradient Descent for Non-Homogeneous Two-Layer Networks: Margin Improvement and Fast Optimization*
  - *Implicit Bias of Gradient Descent for Non-Homogeneous Deep Networks*

  In your related work subsection *“Network embedding principles,”* you only cite a single reference (Zhang et al., 2021). While this is certainly a relevant work, many other studies have considered this classical idea. For instance, you might look at the pruning and lottery ticket hypothesis literature, such as *“Proving the Lottery Ticket Hypothesis: Pruning is All You Need.”*

- **Sections 3 and 4 lack narrative flow.**
  Much of these sections read as a list of definitions, assumptions, theorems, and proofs, rather than a coherent story. There should be a stronger narrative connecting the material back to the main contributions stated in the introduction. Consider deferring some technical details to the appendix to improve readability. See my comments below.

- **The significance of the results is unclear.**
  While there is some attempt to explain their importance in the introduction and conclusion, much more effort is needed to contextualize why these results matter. See my suggestions below for directions to expand on the significance. For example, in the introduction you state:
  > “A fundamental question arises regarding the relationship between solutions found in networks of different sizes.”

But why is this a fundamental question? Does it relate to scaling laws, robustness, generalization, or practical considerations when training larger models?

---

> ### Author Rebuttal · Authors · 2025-07-31
>
> We are grateful to the reviewer for the thorough and insightful feedback on our manuscript. Your comments on the related work, narrative structure, and significance of our results have been instrumental in strengthening the paper and have also highlighted promising future research directions. We have undertaken a substantial revision based on your detailed guidance.
>
> ---
> ### On Related Work and Significance
>
> **Broadened Context of Related Work**
>
> Following your guidance, we have substantially revised our discussion of the related literature in two key areas:
>
> 1.  **“Optimization dynamics and implicit bias”**: We have updated this section's title. We expanded our discussion to cover recent developments, including the analysis of margin in non-homogeneous models  **(Kunin et al., 2023)**、**(Cai et al. 2025)**, and the connection between KKT conditions and benign overfitting **(Frei et al., 2023)**.
>
> 2.  **“Network embedding principles”**:we have broadened the discussion to better situate our work, citing foundational research on hierarchical structures that is highly relevant to our embedding principles **(Fukumizu & Amari, 2000)**, as well as recent studies on permutation symmetries that connect different global minima **(Şimşek et al., 2021)**. we carefully considered the suggested literature on network pruning and the Lottery Ticket Hypothesis. We agree this is an important field, but we concluded that its core principle is conceptually distinct from our work. Specifically, pruning focuses on **identifying and removing** less important neurons to create a sparse sub-network. In contrast, our method focuses on merging neurons that have a specific structural relationship to create a smaller, equivalent network.
>
> **Articulating the Significance of Our Work**
>
> Our contribution is best understood in two interconnected parts:
> 1.  **A Static KKT Point Embedding Principle:** This proves that the solutions (KKT points) of a smaller network's max-margin problem are isometrically embedded within the solution set of any wider network derived from it.
> 2.  **A Dynamic Trajectory Embedding Principle:** This shows that the entire gradient flow training trajectory is similarly preserved under our transformation. The static principle is the essential foundation that guarantees this dynamic preservation.
>
> While the invariance of the network function under neuron splitting may seem intuitive, the core of our contribution lies in rigorously proving that this intuition extends to the optimization landscape's critical points (the KKT points) and, crucially, to the training dynamics (the gradient flow trajectories). This formalization yields significant insights into why overparameterized models can generalize well.
>
> A fundamental question in deep learning is how a highly complex neural network can effectively learn a low-complexity classification boundary. Our work provides a direct theoretical answer:
> * Our KKT embedding principle formally **guarantees the existence of low-complexity solutions** (those embedded from smaller, simpler models) within the vast solution space of overparameterized networks. Prior work has shown that gradient flow converges in direction to a KKT point of the max-margin problem (**Lyu & Li, 2019**), but a significant gap remained: large networks can possess many KKT points with varying complexities. Our work bridges this gap by proving that simple, generalizable solutions are structurally preserved as network width increases.
> * This has immediate practical implications. The co-existence of both low-complexity (embedded) and high-complexity KKT points suggests that **initialization and hyperparameter tuning play a critical role in selecting a solution**. Our framework provides a theoretical basis for understanding this selection process and for future work on designing training procedures that explicitly steer optimizers toward these desirable, low-complexity solutions to improve generalization.
>
> ---
> ### On Narrative Flow and Major Questions
>
> **On the Narrative Flow of Sections 3 and 4**
>
> Following your valuable suggestions, we have improved the narrative flow by:
> * Deferring technical definitions (formerly Def. 3.1, 3.2, Asm. 3.3) to the appendix.
> * Splitting Section 4 into two separate sections: one establishing the general KKT embedding framework and another dedicated to its application to neuron splitting and channel splitting.
>
> ### On the Major Questions
>
> **1. On the relation between Theorem 4.6 and 4.9:**
>  We agree that the two-layer network transformation is a special case of the deep network one. To streamline the paper, we have removed the separate theorem for two-layer networks and now present only the general deep network result as our main theorem. The two-layer construction is now included as a pedagogical example in the appendix.
>
>
> **2. On extending the framework to CNNs:** We have formally extended our framework to homogeneous Convolutional Neural Networks (CNNs) by generalizing "neuron splitting" to "channel splitting" ans included this as a new main result. The transformation splits a channel into $m_{split}$ new channels using coefficients $\{c_i\}$ that satisfy $\sum_{i=1}^{m_{split}} c_i^2 = 1$. It is defined as follows:
>  * **Layer k**: Replacing the filter $F_j^{(k)}$ that produces channel $j$ with $m_{split}$ new filters, where the $i$-th new filter is defined as $c_i F_j^{(k)}$.
>  * **Layer k+1**: For every filter $F_p^{(k+1)}$, replacing its $j$-th input channel slice with $m_{split}$ new slices, where the $i$-th new slice is scaled by the corresponding coefficient $c_i$.
>
>   Successfully proving that this general transformation satisfies the conditions of our core theory (**Theorem 4.5**) is a powerful validation of our framework. It shows that the principles of isometric and output/subgradient preservation are not confined to simple network topologies but provide a robust blueprint for analyzing a wider class of homogeneous networks. This strongly suggests that our framework can be adapted to other architectures with specific structural invariances.
>
> **3. On mapping KKT points from larger to smaller networks:**
> Applying our framework to the larger-to-smaller network mapping via neuron merging is a valuable direction that connects our work to network pruning.
>
> To map Karush-Kuhn-Tucker (KKT) points from a larger network (parameters $\eta$) to a smaller one ($\theta$), we can define a "neuron merging" operator $M$ where $\theta = M(\eta)$. For this operator to be KKT-preserving , it must satisfy conditions analogous to our **Theorem 4.5**, namely:
> 1.  **Output Preservation**: $\Phi(M(\eta); x) = \tilde{\Phi}(\eta; x)$. This is analogous to the output-preserving condition for splitting.
> 2.  **Subgradient Preservation**: $\partial_{\theta}^{\circ}\Phi(M(\eta); x)$ must maintain a specific relationship with $M(\partial_{\eta}^{\circ}\tilde{\Phi}(\eta; x))$, analogous to the subgradient condition for splitting.
>
> The output preservation condition imposes a strict structural constraint. For instance, merging several neurons into one ($\sum_{i}a_i\sigma(b_i^T x) \to a\sigma(b^T x)$) requires the weight vectors $\{b_i\}$ of the neurons to be **collinear**.
>
> While this requirement seems restrictive, it connects to the practical phenomenon of neural condensation, where neuron weights are observed to align and become collinear during training (Zhou, et al.,2022). This suggests that valid opportunities for such a theoretically-grounded merge may arise naturally.
>
> Our future work will explore the precise KKT point mapping from larger to smaller networks under valid merging operations.
>
>
> ### Response to Minor Comments
>
> We have incorporated all of your minor comments:
>
> * Figure 1 has been resized and given a more detailed caption, and it is now referenced at the end of the introduction to serve as a roadmap.
>
> * The technical definitions have been moved to the appendix.
>
> * Section 4 has been split into two separate sections.
>
> * We have clarified the definition of ω-limit sets and added a formal citation.
>
> ---
>
> Once again, we would like to express our gratitude for your thorough and constructive review. Your comments have been instrumental in helping us strengthen the manuscript.
> ### References
>
> * **Kunin, D., et al. (2023).** The Asymmetric Maximum Margin Bias of Quasi-Homogeneous Neural Networks. *International Conference on Learning Representations (ICLR)*.
> * **Cai, Y., et al. (2025).** Implicit Bias of Gradient Descent for Non-Homogeneous Deep Networks. *Proceedings of the 42nd International Conference on Machine Learning*.
> * **Frei, S., et al. (2023).** Benign Overfitting in Linear Classifiers and Leaky ReLU Networks from KKT Conditions for Margin Maximization. *arXiv preprint arXiv:2303.01462*.
> * **Fukumizu, K., & Amari, S. (2000).** Local minima and plateaus in hierarchical structures of multilayer perceptrons. *Neural Networks*.
> * **Şimşek, B., et al. (2021).** Geometry of the Loss Landscape in Overparameterized Neural Networks: Symmetries and Invariances. *Proceedings of the 38th International Conference on Machine Learning (ICML)*.
> * **Lyu, K., & Li, J. (2019).** Gradient descent maximizes the margin of homogeneous neural networks. *arXiv preprint arXiv:1906.05890*.
> * **Zhou, H., et al. (2022).** Towards understanding the condensation of neural networks at initial training. *Advances in Neural Information Processing Systems, 35*.
> * **Hirsch, M. W., Smale, S., & Devaney, R. L. (2013).** *Differential equations, dynamical systems, and an introduction to chaos*. Academic Press.

---

> > ### Comment · Reviewer_rUJh · 2025-08-05
> >
> > Thank you for the detailed response to my review. You have addressed most of my comments and clarified several aspects of your work. My two main concerns were:
> >
> > (1) Related work — In the original submission, the discussion of prior literature was too narrow and omitted several important works. Based on your response, this concern has been largely alleviated.
> >
> > (2) Significance — While your response clarified the theoretical contribution, I still find the significance of the results to be limited. The existence of low-complexity embedded KKT points is now well-formalized, but it remains unclear whether these solutions are relevant in practice. For instance, are such low-complexity solutions actually found by standard training dynamics? Demonstrating this empirically—or providing a stronger argument or even discussion for how initialization (e.g., scale) and optimization hyperparameters (e.g., learning rate) influence the complexity of the converged KKT point—would significantly strengthen the case for the relevance of your results.
> >
> > Overall, I find this work to be technically solid, but it still lacks sufficient contextualization within the broader literature to establish the significance of its theoretical findings. I strongly agree with Reviewer dAiL that the work would benefit from moving some of the technical theory to the appendix and including empirical results to support and motivate the theory. I’ve read your response to that reviewer, and I’d like to emphasize that empirical results should not merely validate the robustness and generality of the theory, but more importantly help contextualize its significance.
> >
> > Based on these points, I will keep my original evaluation of 4. That said, I believe this paper would be more impactful with a rewrite and resubmit.

---

> > > ### Author Response · Authors · 2025-08-09
> > >
> > > Thank you again for your insightful and highly constructive feedback. We value your acknowledgment that our work is "technically solid" and we fully agree with your core point: providing empirical support is the critical next step to establish the significance and practical relevance of our theoretical findings.
> > >
> > > We believe our current work provides the essential, rigorous foundation upon which such an empirical investigation can be built. It formalizes the existence of these low-complexity points, thus providing the precise theoretical framework needed to now ask the empirical questions you've raised.
> > >
> > > The rigorous empirical study you suggest—including verifying if these solutions are found by standard training dynamics and exploring the influence of hyperparameters—is indeed the right way to bridge this gap. As you might anticipate, such a meaningful investigation requires careful design and execution that goes beyond simple validation and is unfortunately not feasible to complete within the short rebuttal period.
> > >
> > > Your review has provided a clear and valuable roadmap for strengthening the impact of this research. We are genuinely grateful for this guidance and are committed to undertaking the empirical work needed to fully realize the potential of these theoretical findings. Thank you again for your invaluable engagement, which will have a profound impact on our subsequent research.

---

### Official Review · Reviewer_jDVY · 2025-07-06

**Clarity:** 2
**Significance:** 2
**Originality:** 3
**Rating:** 4
**Confidence:** 2

**Summary:**

I am very much not an expert in the mathematical side of things here. My understanding is that the paper proves a neat theoretical fact about bias-free ReLU networks. If you take a trained narrow network and simply clone each neuron (splitting one neuron into two identical ones and rescaling weights), every optimal solution and even the whole gradient-flow training path of the small net (under very strict assumptions on the path and the net) sits perfectly inside the bigger net. => widening in this very specific way does not create new, better optima. The authors first show a general optimization theorem and then apply it to two-layer and deep ReLU models.

**Questions:**

- Bias terms: how hard would it be to make this work for NNs with bias?
- Weight sharing: the same question, this would make it much more actionable as a result

My main ask = empirical demo
A tiny experiment on a 2-layer ReLU network would greatly strengthen the paper and I'm primarily interested to see how the theory and the experiment deviate as the conditions in the experiments are not matching exactly the idealized conditions in the theory.

My score would increase if you include convincing experiments.

**Ethical Concerns:**

["NO or VERY MINOR ethics concerns only"]

**Final Justification:**

I am very happy with the additional experiments the authors ran and the way they addressed my concerns. If these are incorporated in the paper, I believe it would definitely be stronger than the original. I am increasing my rating from 3 = borderline reject to 4 = borderline accept. Not being an expert in the sub-field, I'm not confident to do 5 = accept, but I think the paper would be a valuable addition to the field. Thank you for addressing my concerns and running additional experiments!

**Limitations:**

Yes, pretty well addressed modulo the experiment I would like to see.

**Paper Formatting Concerns:**

There are broken links e.g.
Theorem ??
Lemma ??

**Quality:**

3

**Strengths And Weaknesses:**

Strengths:
- Solid proofs: the math seems self-contained and careful (as far as i can tell)
- Clear structure: the paper moves from a general theorem to concrete neural net corollaries, which is easy to follow
- new angle: It is the first work (as far as i can tell) to show that neuron-splitting preserves both KKT optima and whole gradient-flow paths. previous works focused primarily on optima alone

Weaknesses:
- Very strong assumptions: No biases, full batch gradient flow, and perfectly separable data = unrealistic for modern training, and it is therefore hard to say how much of this generalizes to anything real
- No experiments: even a toy example would help me trust the theory more and I would love to see how much the theory deviates from the simple experiments as I make the conditions imperfect
- small presentation issues: several “Theorem ??” placeholders
- Limited scope: results do not cover CNNs, attention layers, or mini-batch SGD (no weight tying permitted in the theory)

---

> ### Author Rebuttal · Authors · 2025-07-31
>
> We are very grateful for your insightful review and constructive feedback. We especially appreciate that you recognized the novelty of our work in showing the preservation of the **entire gradient-flow path**, which is a core part of our contribution. Your feedback has been instrumental in helping us bridge the gap between our theory and its practical implications.
>
> ---
> ### Our Main Response: A Comprehensive Empirical Demonstration
>
> In direct response to your excellent suggestion to bridge the gap between our idealized theory and practice, we have conducted a comprehensive suite of experiments to systematically test the robustness and generality of our Trajectory Embedding Principle. We present these findings as a three-step narrative, moving from ideal conditions to more realistic scenarios. Our theory predicts that the maximum deviation between the two trajectories, $\max_{k \in \{0, 1, \dots, N\}} ||\eta_k - T\theta_k||_2$, will be negligible.
>
> **Step 1 & 2: Verification and Robustness on Toy Models**
>
> We began by verifying our theory under ideal conditions and then tested its robustness when key assumptions were relaxed.
>
> * **Setup:** We constructed 2-layer homogeneous MLPs (with a hidden layer of 1 neuron vs. 2 neurons) and simple 2-layer CNNs (consisting of one convolutional of 2 channels vs. 4 channels and one fully-connected layer). We initialized them such that the initial parameters of the larger network were a direct mapping of the smaller one via our transformation $T$ (i.e., $\eta_0 = T\theta_0$). This transformation T implements a neuron splitting mechanism for MLPs and a channel splitting mechanism for CNNs. The latter is described in detail in our response to Question 2.The latter is described in detail in our response to Question 2. Both networks were then trained under various conditions for 1 million steps on a two-dimensional linearly separable dataset.
>
> * **Conclusion:** The results, summarized below, confirm that the trajectory error remains extremely small across all setups.
>
> | Exp. | Model & Optimizer | Data Condition | Max Trajectory Error  |
> | :--- | :--- | :--- | :--- |
> | 1    | MLP with GD       | Separable      | `4.46e-13`             |
> | 2    | MLP with SGD      | Separable      | `4.35e-13`             |
> | 3    | MLP with GD       | Non-Separable  | `5.57e-13`             |
> | 4    | CNN with GD       | Separable      | `2.36e-10`             |
>
> The small, non-zero errors (on the order of `1e-13` to `1e-10`) are an expected consequence of numerical precision. Over a million steps, minuscule floating-point rounding errors inevitably accumulate.
>
> It is crucial to note that for the SGD experiment (Exp. 2), both networks were trained using an **identical sequence of mini-batches**. We also ran a control experiment where the two networks were trained with **different, randomized batch orders**. In this scenario, the maximum trajectory error increased dramatically to **`1.5316e-01`**. This result is expected, as a network's training path is highly dependent on its specific data sequence.
>
> **Step 3: Demonstrating Generality on MNIST**
>
> Finally, to address the scope and demonstrate the framework's applicability to more realistic architectures, we extended our tests to the MNIST dataset.
>
> * **Setup:** We scaled up to the MNIST dataset (classifying 3s vs 5s), training both **two-layer** homogeneous MLPs (128 vs 256 hidden units) and **two-layer** CNNs (10 vs 20 channels). We used SGD with a **learning rate of 0.001 and a batch size of 64** for **1000 epochs**. The models achieved high classification accuracy **(e.g., 99.99% for MLP, 99.79% for CNN)**, ensuring we observed the trajectory error during a meaningful learning phase.
>
> * **Conclusion:** The results, summarized below, confirm that the trajectory preservation principle holds with remarkable precision even on this more complex task. Both the MLP and CNN models maintained a near-zero trajectory error throughout training, demonstrating that the training dynamics of the wider network are highly predictable from its narrower counterpart.
>
> | Model Configuration  | Max Trajectory Error |
> | :---                | :---                 |
> | MLP (Homogeneous) | ` 2.1868e-13`         |
> | CNN (Homogeneous) | `1.2839e-13`         |
>
> This comprehensive suite of experiments confirms that our theoretical principle is not only correct in its ideal setting but is also a robust and generalizable framework for understanding the relationship between homogeneous networks of different sizes.
>
> We will add a dedicated section for these results to the main paper. A detailed description of all experimental setup will be provided in a dedicated appendix section in our revised manuscript. Furthermore, we will make the code publicly available on GitHub to ensure full reproducibility.
>
> ---
> ### On Assumptions, Scope, and Specific Questions
>
> **On Strong Assumptions and Limited Scope:** We agree that our theoretical setup involves strong assumptions. However, we wish to clarify a critical distinction: the assumptions of full-batch gradient flow and perfectly separable data are not required for our KKT and Trajectory Embedding Principles.
>
> These conditions are primarily needed for the well-known result from prior work that guarantees the training direction converges to a max-margin KKT point. In contrast, our embedding principles (Theorems 4.5 and 5.2) describe a more fundamental structural relationship between networks. This relationship holds regardless of whether the training converges to that specific max-margin solution.
>
> Our new experiments explicitly confirm this. The trajectory embedding holds with near-perfect precision even when using Stochastic Gradient Descent (Exp. 2 & MNIST) and on non-separable data (Exp. 3). This demonstrates that our framework is robust and not confined to those idealized convergence scenarios. Furthermore, to broaden the scope, we are happy to clarify that our revised manuscript introduces a new main result extending our framework to Convolutional Neural Networks (CNNs).
>
> **Question 1: How hard would it be to include bias terms?**
> This is a challenging but important question. Extending our framework to include bias terms is non-trivial because biases break the network's strict homogeneity, which is a cornerstone of our current analysis and the associated max-margin dynamics. Specifically, this extension (with bias) would require a fundamental redesign of the corresponding linear transformation T, which currently relies on this homogeneous property.
>
> However, very recent work has begun to bridge this gap. For instance, Cai et al. (2025) extend the implicit bias analysis to a broad class of near-homogeneous networks. Their framework is powerful because it formally includes architectures with components that break strict homogeneity, such as bias terms and residual connections. They show that for these networks, gradient descent still converges in direction to a KKT point of a corresponding margin maximization problem. This development provides a promising theoretical pathway for our own work. A natural future direction is to investigate whether our KKT embedding principle can be generalized to this near-homogeneous setting, thereby extending our results to practical networks that include biases.
>
> **Question 2: How hard would it be to handle weight sharing?**
>
> To demonstrate the generality and power of our approach, **we have introduced a new main result extending our principle to Convolutional Neural Networks (CNNs)**.This required designing a novel **'channel-splitting'** transformation, meticulously constructed to respect the architectural hallmarks of CNNs, such as weight sharing and locality. The transformation isotropically splits a single feature channel into $m_{split}$ new channels via coefficients $\{c_i\}$ that satisfy $\sum_{i=1}^{m_{split}} c_i^2 = 1$. The specific construction is as follows:
>
> * **At Layer k (Outgoing Weights):** The filter $F_j^{(k)}$ responsible for generating channel $j$ is replaced by $m_{split}$ new filters. The i-th new filter is simply a scaled version of the original: $c_i F_j^{(k)}$.
> * **At Layer k+1 (Incoming Weights):** Consequently, every filter in the next layer that receives channel $j$` as input has its j-th input slice replaced by $m_{split}$ new slices, with each new slice being scaled by the corresponding coefficient $c_i$.
>
> Successfully proving that this transformation satisfies our core theory (**Theorem 4.5**) validates our framework as a flexible blueprint applicable to a broad class of homogeneous networks, with future work poised to explore architectures like Transformers.
>
> ---
> ### Minor Issues
>
> The "Theorem ??" placeholders were indeed artifacts from our drafting process and have all been corrected in the final version. We apologize for the oversight.
>
> ---
>
> Thank you again for your valuable and pragmatic feedback. We believe the new empirical demonstrations, a broader scope that now includes CNNs, and our detailed clarifications directly address your main concerns. We hope these additions will convince you of the paper's value and merit your support.
>
> ### References
>
> * Cai, Y., et al. (2025). Implicit Bias of Gradient Descent for Non-Homogeneous Deep Networks. *Proceedings of the 42nd International Conference on Machine Learning*.

---

> > ### Comment · Area_Chair_RRjD · 2025-08-07
> > **IMPORTANT REMINDER: PLEASE PARTICIPATE IN THE REBUTTAL**
> >
> > Dear Reviewer,
> >
> > I would like to remind you that reviewers MUST participate in discussions with authors.
> > If you don not engage, your review will be flagged as insufficient, and you have to live
> > with the consequences.
> >
> > best
> > your ac

---

> ### Comment · Reviewer_jDVY · 2025-08-09
> **Score increase**
>
> Thank you for your detailed experiments and revisions. I am increasing my score. Good luck!

---

> > ### Author Response · Authors · 2025-08-09
> >
> > Thank you so much for raising your score. We are particularly pleased that our new experiments and revisions successfully addressed your initial concerns. Your feedback was instrumental in strengthening the paper, and we are very grateful.

---

### Official Review · Reviewer_ZZqy · 2025-07-20

**Clarity:** 4
**Significance:** 1
**Originality:** 2
**Rating:** 3
**Confidence:** 4

**Summary:**

A mathematical paper shows that a particular neuron splitting operation in homogeneous neural networks (i.e. neural networks f(x, \theta) with input x and parameters \theta, such that f(x, c \theta) = c^L f(x, \theta) for some L; e.g. networks without biases, and with ReLU or linear activation functions), preserves Karush-Kuhn-Tucker (KKT) conditions and gradient flow dynamics. The non-differentiability of ReLU at 0 is addressed using Clarke's subdifferential.

**Questions:**

- line 197: What is "(Theorem 4.1 in your original numbering)" referring to?

**Ethical Concerns:**

["NO or VERY MINOR ethics concerns only"]

**Final Justification:**

Thank you for your detailed and thoughtful response. I apologize for calling the formal details a good exercise; I realize that may come across as rude. I am a reader who is comfortable with mathematical formalism but prefers learning about new ideas succinctly and intuitively, leaving technical details to a nicely written appendix.

From this perspective, I disagree that the level of formalization yields significant insights into why overparameterized models can generalize well. For me, the key takeaway remains point 2 of Theorem 4.6: that transformation T preserves the gradient for homogeneous networks. This is an interesting result, but of limited scope due to the homogeneity assumption. This result can be presented straightforwardly without using Clarke's subdifferential. Of course, it is worth mentioning the consequences of this insight for deeper networks, convolutions, KKT conditions, gradient dynamics, and generalization; however, these insights can be conveyed intuitively and much more succinctly.

I think your revision goes in this direction, but without having read a substantially revised version, I prefer not to vote for acceptance.

**Limitations:**

See significance and originality.

**Quality:**

3

**Strengths And Weaknesses:**

**Quality**

- The paper is carefully written.

**Clarity**

- The paper is very clearly written. I read it linearly and could always follow.

**Significance**

- The scope (a specific kind of neuron splitting in homogeneous neural networks) is very limited.

- While I appreciate the clarity of the formal details, the result that this neuron splitting operation leaves KKT conditions and gradient flow invariant is intuitive and could be explained much more succinctly. Many formal details feel to me more like good exercises for students getting familiar with subdifferentials and KKT conditions.

**Originality**

- Neuron splitting has been studied in many other works. I haven't come across the specific splitting operation T(a, b) = (c_1 a, ..., c_k a, c_1 b, ..., c_k b) needed in this paper to assure subdifferential preservation.

- The propositions and theorems look original to me, but the proof techniques are straightforward.

---

> ### Author Rebuttal · Authors · 2025-07-30
>
> We sincerely thank the reviewer for their positive feedback on the clarity and careful writing of our manuscript. We appreciate the thoughtful critique and would like to address the primary concerns regarding the paper's significance and originality, which we believe are more substantial than perceived.
>
> ---
> ### On the Significance and Originality of Our Work
>
> We respectfully argue that the significance of our findings is substantial, as it provides a foundational, structural explanation for key questions in deep learning generalization. Our contribution is best understood in two interconnected parts:
> 1.  **A Static KKT Point Embedding Principle:** This proves that the solutions (KKT points) of a smaller network's max-margin problem are isometrically embedded within the solution set of any wider network derived from it.
> 2.  **A Dynamic Trajectory Embedding Principle:** This shows that the entire gradient flow training trajectory is similarly preserved under our transformation. The static principle is the essential foundation that guarantees this dynamic preservation.
>
> While the invariance of the network function under neuron splitting may seem intuitive, the core of our contribution lies in rigorously proving that this intuition extends to the optimization landscape's critical points (the KKT points) and, crucially, to the training dynamics (the gradient flow trajectories). This formalization is not merely a "good exercise"; it yields significant insights into why overparameterized models can generalize well.
>
> A fundamental question in deep learning is how a highly complex neural network can effectively learn a low-complexity classification boundary. Our work provides a direct theoretical answer:
> * Our KKT embedding principle formally **guarantees the existence of low-complexity solutions** (those embedded from smaller, simpler models) within the vast solution space of overparameterized networks.  Prior work has shown that gradient flow converges in direction to a KKT point of the max-margin problem (**Lyu & Li, 2019**), but a significant gap remained: large networks can possess many KKT points with varying complexities. Our work bridges this gap by proving that simple, generalizable solutions are structurally preserved as network width increases.
> * This has immediate practical implications. The co-existence of both low-complexity (embedded) and high-complexity KKT points suggests that **initialization and hyperparameter tuning play a critical role in *selecting* a solution**. Our framework provides a theoretical basis for understanding this selection process and for future work on designing training procedures that explicitly steer optimizers toward these desirable, low-complexity solutions to improve generalization.
>
> **Regarding Originality:** While neuron splitting has been studied, our work's originality lies not in the operation itself, but in the **development of a formal KKT Point Embedding Principle that uses such operations to create a rigorous link between the solutions and training dynamics of networks of different sizes**. We are the first to establish this comprehensive static and dynamic correspondence under a unified theoretical framework.
>
> Regarding the proof techniques, we agree that they are direct applications of established tools like the Clarke subdifferential. We see this as a strength, lending clarity and rigor to our results. The novelty lies not in the techniques themselves, but in their application to establish a new and significant principle that was previously unformalized.
>
> ---
> ### Revisions to Enhance Contribution and Clarity
>
> We appreciate your perspective on the paper's limitations of our core results. To better demonstrate the significance of our framework and the depth of its contributions beyond the initial intuition, we have made the following substantial revisions
>
> **1. Broadened Scope with Extension to CNNs:** We acknowledge that focusing solely on fully-connected homogeneous networks could be perceived as a limitation. To demonstrate the generality and power of our approach, **we have introduced a new main result extending our principle to Convolutional Neural Networks (CNNs)**. This required designing a novel **'channel-splitting'** transformation, meticulously constructed to respect the architectural hallmarks of CNNs, such as weight sharing and locality. The transformation isotropically splits a single feature channel into $m_{split}$ new channels via coefficients $\{c_i\}$ that satisfy $\sum_{i=1}^{m_{split}} c_i^2 = 1$. The specific construction is as follows:
> * **At Layer k**: The filter $F_j^{(k)}$ that produces channel $j$ is replaced by $m_{split}$ new filters, where the i-th new filter is defined as $c_i F_j^{(k)}$.
> * **At Layer k+1**: In turn, every filter in the next layer that takes channel $j$ as input has its j-th input slice replaced by $m_{split}$ new slices, where the i-th new slice is scaled by the corresponding coefficient $c_i$.
>
> Successfully proving that this transformation satisfies our core theory (**Theorem 4.5**) validates our framework as a flexible blueprint applicable to a broad class of homogeneous networks, with future work poised to explore architectures like Transformers.
>
> **2. Improved Narrative and Contribution Density:** To address the impression that the paper was overly detailed or like an "exercise," we have significantly restructured the manuscript for a more focused and impactful narrative. This includes:
> * **Consolidating Theorems:** We have consolidated the application theorems. The two-layer network case, being a special case of the deep network one, has been moved to the appendix, allowing the main text to focus on the more general and powerful deep network result.
> * **Streamlining Presentation:** We have deferred highly technical definitions (formerly Def. 3.1, 3.2, and Asm. 3.3) to the appendix to improve readability and focus on the core contributions.
>
> ---
> ### Response to Minor Issue
>
> * **Comment on line 197:** We sincerely apologize for this oversight. This was a remnant from an internal draft and has been removed in the revised version.
>
> ---
>
> Once again, we thank you for your time and valuable feedback. We hope these clarifications and the significant revisions to the manuscript, including the new CNN result, will help you re-evaluate the significance and originality of our contribution more favorably.
> ### References
>
> * **Lyu, K., & Li, J. (2019).** Gradient descent maximizes the margin of homogeneous neural networks. *arXiv preprint arXiv:1906.05890*.

---

> > ### Comment · Area_Chair_RRjD · 2025-08-07
> > **IMPORTANT REMINDER: PLEASE PARTICIPATE IN THE REBUTTAL**
> >
> > Dear Reviewer,
> >
> > I would like to remind you that reviewers MUST participate in discussions with authors.
> > If you don not engage, your review will be flagged as insufficient, and you have to live
> > with the consequences.
> >
> > best
> > your ac

---

> > ### Comment · Reviewer_ZZqy · 2025-08-07
> > **Thanks for the Rebuttal**
> >
> > Thank you for your detailed and thoughtful response. I apologize for calling the formal details a good exercise; I realize that may come across as rude. I am a reader who is comfortable with mathematical formalism but prefers learning about new ideas succinctly and intuitively, leaving technical details to a nicely written appendix.
> >
> > From this perspective, I disagree that the level of formalization yields significant insights into why overparameterized models can generalize well. For me, the key takeaway remains point 2 of Theorem 4.6: that transformation T preserves the gradient for homogeneous networks. This is an interesting result, but of limited scope due to the homogeneity assumption. This result can be presented straightforwardly without using Clarke's subdifferential. Of course, it is worth mentioning the consequences of this insight for deeper networks, convolutions, KKT conditions, gradient dynamics, and generalization; however, these insights can be conveyed intuitively and much more succinctly.
> >
> > I think your revision goes in this direction, but without having read a substantially revised version, I prefer not to vote for acceptance.

---

> > > ### Author Response · Authors · 2025-08-09
> > >
> > > Thank you very much for your insightful and detailed feedback, and for your continued engagement with our work. We also appreciate the clarification regarding your previous comment; we understood your intention was constructive.
> > >
> > > We are particularly grateful for your profound insights on balancing formal rigor with intuitive explanation. Your perspective on the paper's presentation and structure provides us with a very valuable reference.
> > >
> > > We are pleased that you found our key result—that the transformation T preserves the gradient for homogeneous networks (Theorem 4.6)—to be an interesting insight. This is indeed the central pillar of our argument, and we greatly appreciate you recognizing its value.
> > >
> > > Regarding the use of formalism like Clarke's subdifferential, our intention was to ensure the argument was rigorously grounded, especially given the non-differentiable activations common in modern networks. We understand your concern that complex technical details might obscure the intuitive presentation of the core idea. How to achieve the best balance between rigor and intuition is indeed a very important question and one that merits deep consideration.
> > >
> > > Your feedback is highly illuminating and provides us with an important perspective for reviewing and refining our work from different angles. Thank you again for the valuable time and effort you have dedicated to reviewing our manuscript.

---

### Decision · Program_Chairs · 2025-09-17

**Decision:**

Accept (poster)

**Comment:**

The paper enjoyed a fruitful discussion with some reviewers increasing their score.
Based, on the reviews, the discussion, and my own reading here are the pros and cons:

Strength:
- Interesting approach and, compared to many other NeurIPS papers, a solid mathematical approach
- Theorem 5.2: It which shows that training trajectories are preserved under the considered neuron splitting

Weakness:
- Writing, narrative, and text organization require a substantial overhaul: Even for a mathematical text there is lot room for improvements
- Strong assumptions on architecture: positive homogeneous networks exclude biases and many activation functions

If the writing was better, I would recommend acceptance. In its current form, however, the paper should be compared to
competing papers in the gray zone. As a result of this comparison, the paper got accepted.